# Cdc42 mobility and membrane flows regulate fission yeast cell shape and survival

David M. Rutkowski [1], Vincent Vincenzetti[2], Dimitrios Vavylonis [1] ✉ & Sophie G. Martin [2,3] ✉

Polarized exocytosis induced by local Cdc42 GTPase activity results in membrane flows that deplete low-mobility membrane-associated proteins. A reaction-diffusion particle model comprising Cdc42 positive feedback activation, hydrolysis by GTPase-activating proteins (GAPs), and flow-induced displacement by exo/endocytosis shows that flow-induced depletion of low mobility GAPs promotes polarization. We modified Cdc42 mobility in *Schizosaccharomyces pombe* by replacing its prenylation site with 1, 2 or 3 repeats of the Rit C-terminal membrane-binding domain (ritC), yielding alleles with progressively lower mobility and increased flow-coupling. While Cdc42-1ritC cells are viable and polarized, Cdc42-2ritC polarize poorly and Cdc42-3ritC are inviable, in agreement with model's predictions. Deletion of Cdc42 GAPs restores viability to Cdc42-3ritC cells, verifying the model's prediction that GAP deletion increases Cdc42 activity at the expense of polarization. Our work demonstrates how membrane flows are an integral part of Cdc42-driven pattern formation and require Cdc42-GTP to turn over faster than the surface on which it forms.

Polarized secretion, the directional and selective release of molecules from a cell, is critical in many biological processes, including cell communication, tissue development, and immune response. In fungi and other walled cells, it underlies polarized cell growth, as it is necessary for the local delivery of cell wall remodeling and membrane material[1,2]. Yeast model organisms have been extensively used to understand how the small GTPase Cdc42, which associates with the plasma membrane through a prenyl moiety at its C-terminus, defines the site of polarized growth upon local activation, and dynamically marks the site of polarity[3,4].

The Cdc42 polarity system in yeast is believed to represent one of the best examples of biological pattern formation on cell membranes through reaction and diffusion[5–8], as initially proposed by Alan Turing[9]. This type of pattern formation is driven by non-equilibrium fluxes generated by chemical reactions, combined with spatially varying mobilities (we will use the term "mobility" to indicate both diffusion along the membrane as well as association and

dissociation from the membrane, which we however distinguish where possible in the results below). Yeast cells can undergo spontaneous polarization, or symmetry breaking, in which a polarized system arises from a homogeneous one, in absence of a spatial cue. Cdc42 is locally converted to the active, GTP-bound form by the action of a guanine nucleotide exchange factor (GEF). Activation by GEF implements a positive feedback mechanism, where a scaffold protein (Scd2 in *S. pombe*, Bem1 in *S. cerevisiae*) promotes the formation of a complex between Cdc42-GTP and its GEF, leading to activation of neighboring Cdc42 molecules[10–14]. Local Cdc42 activation and amplification leads to the formation of one or several zones of Cdc42 activity, as well as the consequent accumulation of Cdc42 at sites of activity. Cdc42 accumulation is thought to be driven by the slower mobility of Cdc42-GTP than Cdc42-GDP at the plasma membrane[5,7]. In *S. cerevisiae*, thanks in part to extraction of Cdc42-GDP from the plasma membrane by the Guanine nucleotide Dissociation Inhibitor (GDI), this form exchanges with the cytosolic pool

[1]Department of Physics, Lehigh University, Bethlehem, PA, USA. [2]Department of Fundamental Microbiology, University of Lausanne, Lausanne, Switzerland. [3]Department of Molecular and Cellular Biology, University of Geneva, Quai Ernest-Ansermet 30, Geneva, Switzerland. ✉e-mail: vavylonis@lehigh.edu; sophie.martin@unige.ch

more rapidly than Cdc42-GTP, yielding local Cdc42 enrichment[15,16]. In *S. pombe*, the GDI-dependent cytosol-membrane exchange plays a more modest role, but Cdc42-GTP is nevertheless less mobile than Cdc42-GDP, so local activation traps Cdc42[17].

Several mathematical reaction-diffusion models have described Cdc42 polarization mechanistically. A foundational model, which belongs to the prototypical activator–substrate type[18], describes a slowly mobile Cdc42-GTP activator undergoing autocatalytic amplification through the scaffold-mediated feedback described above, fed by the more mobile Cdc42-GDP substrate[19]. Subsequent models, all containing critical non-linear amplification of Cdc42-GTP, have for instance explored the role of the GDI[16,20], of negative feedback[21] and of competition between polarity sites[22] to generate a robust polarity site. However, these models have only implicitly considered the role of GTPase Activating Proteins (GAPs), which are important inhibitory components of the Cdc42 reaction cycle and which are themselves non-uniformly distributed across the cell, to Cdc42 pattern formation. To account for these non-uniform distributions, a model postulated that GAPs switch between fast and slow diffusing states[23].

With a few exceptions, most modeling work has not explicitly considered the role of the membrane as an evolving element in the Cdc42 system. Consideration of membrane flux is however critical, as the direct outcome of Cdc42 local activity is to promote polarized secretion[24,25]. Secretory vesicles bring not only material for cell wall remodeling, which, in walled cells, is essential for local expansion of the cell envelope, but also new plasma membrane, which is largely recycled through endocytosis around the zone of secretion. In principle, this membrane flux can have two opposite effects: vesicles can act as delivery vehicles to bring proteins to the plasma membrane[26]; however, if protein concentration on the vesicle is lower than that at the plasma membrane, they can conversely have a dilution effect[27,28].

We recently showed that polarized secretion, coupled with membrane retrieval by endocytosis in a wider zone, leads to the production of in-plane bulk plasma membrane flows away from the zone of membrane insertion[29] (Figures S1A, S1B). These flows have important consequences on the distribution of membrane-associated proteins, which depends on their mobility. In a model of flows from a fixed zone of secretion[29], long-lived proteins at the plasma membrane are transported by the flow and diluted from the secretion zone. Specifically, for membrane flows with velocity $v$ ($\approx 3$ nm/s for *S. pombe*) over length scale $L$ ($\approx 2\,\mu$m for *S. pombe*), proteins remain bound to the membrane during transport when $L < v/k_{\text{off}}$, where $k_{\text{off}}$ is the dissociation rate. Diffusion with diffusion coefficient $D$ will cover a smaller distance compared to such transport when $D/v < L$. Thus, membrane-associated proteins whose lateral diffusion and dissociation rates are slow enough to satisfy $D/v < L < v/k_{\text{off}}$ are displaced from the secretion zone, as they are unable to dynamically repopulate the naked membrane at the site of secretion[30]. For instance, the fast-exchanging small membrane-associated peptide ritC (the C-terminal membrane-binding peptide of the mammalian Rit GTPase[31];), which exhibits a uniform distribution at the plasma membrane, becomes depleted from secretion zones upon reduction of its exchange dynamics through tandem trimerization[29] (Figure S1A).

In-plane membrane flows can directly influence cell patterning by modulating the distribution of Cdc42 regulators. In fission yeast, three GAPs promote Cdc42-GTP hydrolysis: Rga4 and Rga6 are depleted from cell poles, at least in part due to membrane flows, and accumulate at cell sides[29,32–34] (Figure S1A). At cell sides, Rga4 and Rga6 actively antagonize Cdc42 activation[33–36]. A third GAP, Rga3, co-localizes with Cdc42-GTP zones, but functions additively to the two others, such that triple mutant cells display excessive Cdc42 activity and zone sizes, leading to round cells[36]. We previously experimentally demonstrated the importance of flows in cell patterning by constructing an optogenetically-controlled GAP (optoGAP) that couples to membrane flows only in the light. We showed that flow-induced depletion of the

Cdc42 GAP from zones of active secretion is necessary and sufficient to promote cell polarization[29] (Figure S1A). Displacement of the GAP to the edge of the secretion zone leaves a zone permissive for Cdc42 activity and corrals it, promoting the formation of the polarity site, and defining its size and the width of the cell. Thus, physical, non-equilibrium membrane fluxes can in principle provide a driving force for pattern formation, in addition to the classic non-equilibrium fluxes driven by reaction and diffusion.

Membrane fluxes may influence the Cdc42 protein itself. Cdc42 prenylation occurs in the ER and prenylated proteins are delivered to the plasma membrane by membrane trafficking[37], but Cdc42 also exchanges between membrane and cytosolic fractions. While polarized delivery of Cdc42 on secretory vesicles had been initially proposed as the main positive force behind Cdc42 polarization[26], this is not required for, nor can explain, the accumulation of Cdc42 at sites of polarity[27,28,38]. Replacement of the Cdc42 prenylation site by the ritC peptide, which drives Cdc42 membrane binding directly from the cytosol, also demonstrated that delivery by secretory vesicles is not important for Cdc42 polarization in either budding or fission yeast, as this Cdc42 variant is largely functional and accumulates at sites of activity[17]. Instead, in models that explicitly considered membranes, membrane delivery was predicted to weaken the polarity site[27,28] or promote its displacement during pheromone gradient tracking[39]. Another study proposed that Cdc42 removal by an endocytic coral around a site of exocytic Cdc42 delivery is important for robust polarity, without considering its displacement along the plasma membrane by membrane flow[40]. The effect of in-plane membrane flows on Cdc42 symmetry breaking—especially as Cdc42 activity itself defines the flow pattern—has not been addressed.

In this work, we probe how Cdc42 molecules, whose activity establishes the flows, are themselves affected by membrane flows, i.e., we do not simply consider flows as a downstream consequence of local Cdc42 activity, but as an element that feeds back to Cdc42, by modifying the distribution of both Cdc42 itself and its negative GAP regulators. We present a reaction-diffusion particle-based model of Cdc42 subject to exo-/endocytosis-induced membrane flows, which shows that robust polarization relies on flow-induced depletion of a low-mobility GAP. We then experimentally reduce Cdc42 mobility at the membrane and demonstrate, in agreement with model predictions, that 1) Cdc42 reduced mobility correlates with expansion of the active zone, and 2) strong mobility reduction is incompatible with viability due to GAP-mediated inactivation. Indeed, GAP depletion increases Cdc42 activity, restoring viability at the expense of loss of polarization. Our combined modeling and experimental studies demonstrate how membrane flows are an integral part of the Cdc42-driven pattern formation. They provide non-intuitive evidence that the high mobility of Cdc42 GTPase is a key ingredient of its polarized distribution.

## Results

### Reaction-diffusion particle-based model of Cdc42 with membrane flows

We developed a particle-based model of Cdc42 patch formation which includes the effects of membrane flow together with a reaction-diffusion mechanism for Cdc42 activation and mobility (Fig. 1A–C, Materials and Methods). Representing each protein or protein complex as a particle allows a direct implementation of transport by membrane flow[29] as well as accounting for finite-number concentration fluctuations[41]. The chemical reaction scheme (Fig. 1A, Table S1) accounts for Cdc42-GDP and Cdc42-GTP membrane diffusion as well as binding and unbinding to the plasma membrane from a cytoplasmic pool of uniform particle concentration. The Scd1/Scd2 GEF complex associates to the plasma membrane or to Cdc42-GTP on the membrane where it catalyzes the conversion of nearby Cdc42-GDP to Cdc42-GTP. This positive feedback retains the essential components

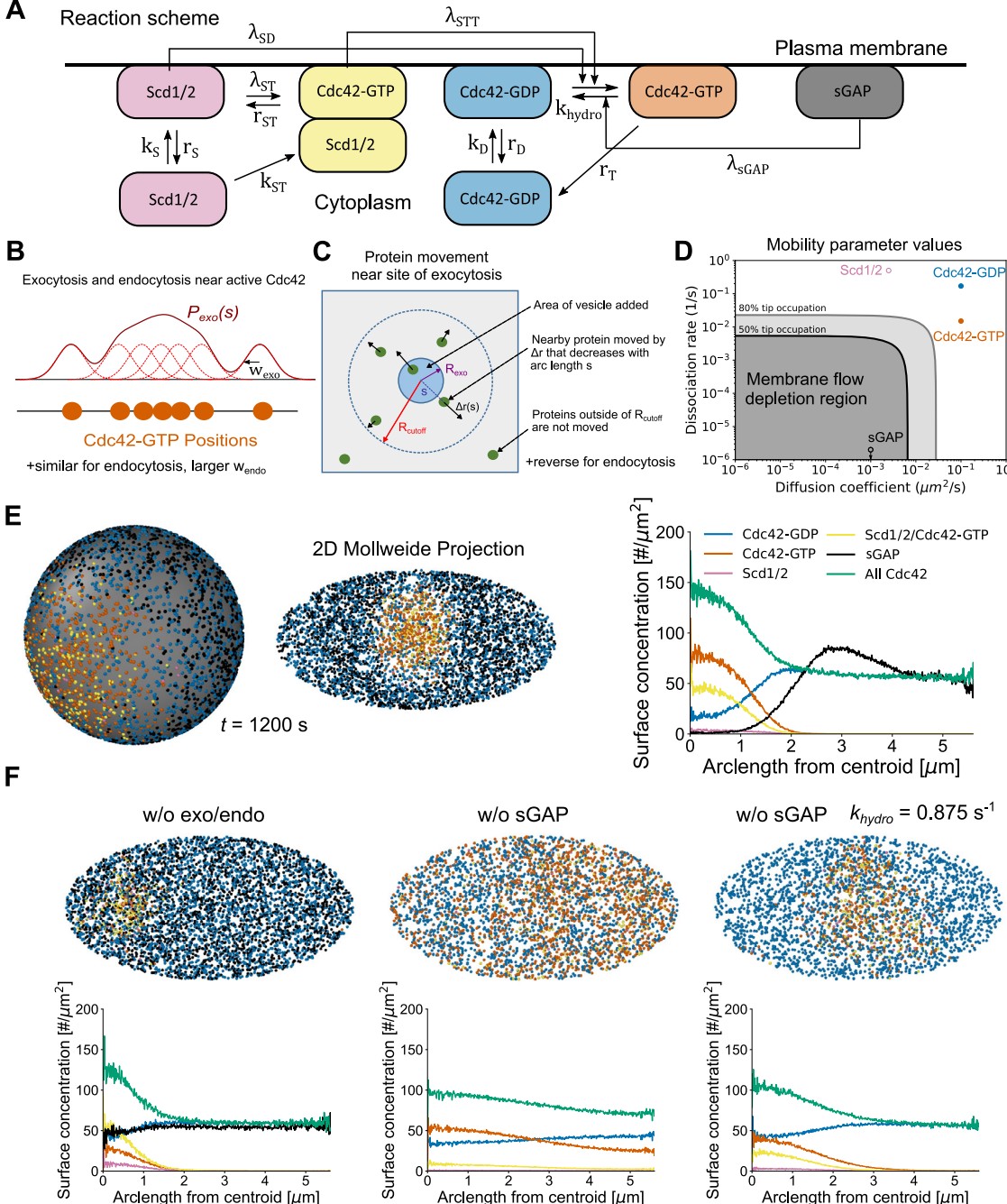

**Fig. 1 | Computational model of Cdc42 polarization in the presence of secretion-driven membrane flows. A** Reaction scheme on cell membrane (horizontal line). Cdc42-GTP is formed by two reactions mediated by either Scd1/2 or by Scd1/2/Cdc42-GTP. Boxes not attached to membrane are cytoplasmic. **B** Exocytosis and endocytosis probability distributions are the convolution of the Cdc42-GTP particle positions and a Gaussian function. **C** Exocytosis (endocytosis) instantaneously pushes (pulls) particles away from (towards) the site of the event by an amount $\Delta r(s)$ that depends on their arclength distance from the site of the event. $R_{cutoff}$ indicates the maximum range of exo/endocytosis. From[29] ©The Authors, some rights reserved; exclusive licensee AAAS. Distributed under a Creative Commons Attribution NonCommercial License 4.0 (CC BY-NC) http://creativecommons.org/licenses/by-nc/4.0/ **D** Reference model mobility parameters. Solid gray and black lines mark boundaries where inert proteins are moderately (80%) or strongly (50%) affected by simulated membranes flows of WT exo/endocytosis centered towards a point of the spherical simulation domain (Figure S1B). Tip occupation ratio is defined as the density of particles near the tip region ($<\pi R/4$ arclength away from top tip) divided by the density in the remaining area of the sphere[29]. The mobilities of Cdc42, Scd1/2 and sGAP are indicated. **E** Patch formation occurs for reference case parameters seen from both the 3D sphere, 2D Mollweide projection, and surface concentration profiles from the patch centroid. **F** Patch formation is weakened at reference parameters by either turning off exo/endocytosis (left) or removing the discrete GAP particle, sGAP (middle). By increasing the hydrolysis rate of Cdc42-GTP from 0.175 /s to 0.875 /s, a more distinct patch is recovered without sGAP (right). Snapshots were taken at 1200 s. For graphs in (**D**, **E**, **F**), source data are provided as a Source Data file.

needed for active patch formation, following a prior particle model for *S. cerevisiae*[41]. Slowly diffusing GAPs that catalyze the hydrolysis of nearby Cdc42-GDP are also included as discrete particles. These GAP particles represent GAP proteins with long membrane residence time, which are thus displaced by membrane flows, such as the optoGAP described above or the endogenous proteins Rga4 and Rga6, which localize to the cell sides[29,32–34]. Therefore, we labeled these discrete GAP particles as sGAPs (slow GAPs). The effect of Rga3, which localizes

at cell tips together with Cdc42-GTP[36] is implicitly included in the rate of hydrolysis $k_{hydro}$. We do not attempt to describe the competition for Cdc42 between the two tips of *S. pombe*[42]; instead we focus on a simple representation of a single tip where we implemented the reactions on a spherical or spherocylindrical domain. The number of particles in the simulations is based on measured protein or mRNA concentrations. All rates are assumed uniform across the sphere, so patterns can only arise via symmetry breaking.

Membrane-associated proteins are displaced by vesicle-driven membrane flows[29]. In cells, the sites of exo- and endocytosis are defined by local Cdc42 activity. To reproduce this in simulations, exo- and endocytic events, modeled as delivered or removed individual patches of membrane that displace the protein particles at the membrane, occur nearby active Cdc42 (Fig. 1B). Specifically, the spatial probability distributions for exocytosis and endocytosis are convolutions of Cdc42-GTP particle positions with a Gaussian of width $w_{exo}$ and $w_{endo}$, which we calculated by comparing tip distributions of CRIB-GFP (marking Cdc42-GTP) to profiles of exocytosis and endocytosis in wild type cells[17,29]. Individual delivery or removal sites were picked from this distribution with a rate matching the corresponding rates of endocytosis and exocytosis at cell tips[29]. Particles nearby sites of exocytosis (endocytosis) were moved radially inwards (outwards) and instantaneously by $\Delta r(s)$, a decreasing function of arc-length distance $s$ to the site (Fig. 1C). The displacement $\Delta r(s)$ is based on a model accounting for area conservation of a tensed membrane and is parametrized by two coefficients of order unity describing the fraction of mobile membrane and ratio of lipid to protein velocity[29]. Membrane flux is due in part to the width of the exocytosis profile being narrower than that of endocytosis[29], and in part to the higher rate of exocytosis than endocytosis ($r_{exo}A_{exo} = 0.0214\ \mu m^2/s$ while $r_{endo}A_{endo} = 0.0157\ \mu m^2/s$, based on experimental estimates[29]) leading to net cell growth. For simplicity we report results for flows on the surface of a sphere without explicitly simulating growth, assuming the center of a Cdc42 patch corresponds to the frame of reference of a growing tip. We checked that simulations on a growing or static spherocylinder domain with particle movements away from the tip set to be the same as the average membrane flows of our reference sphere simulations lead to very similar results (Figure S2).

As a result of membrane flows, particles with slow dissociation rate and diffusion coefficients lying within a depletion region are displaced away from a cell tip when using stationary polarized wild type profiles for endocytosis and exocytosis (Fig. 1D, S1B)[29]. The model assumes that Scd1/2 rate constants lie well outside the depletion region, similar to fast kinetics of the Bem1 complex in budding yeast[41], while sGAP lies within the depletion region (Fig. 1D). The dissociation rates and membrane diffusion coefficients of Cdc42-GDP and Cdc42-GTP also lie outside of the depletion region (Fig. 1D). Cdc42-GDP rates were estimated from fluorescence recovery after photobleaching (FRAP) experiments of a functional tagged form of Cdc42 expressed at the native locus, Cdc42-mCherry[SW 17]. Specifically, we performed fits of FRAP experiments at cell sides, which we assume is dominated by Cdc42-GDP kinetics (Figures S3A–C). To resolve, as best as possible, lateral diffusion versus membrane detachment rates, we performed and analyzed FRAP for bleaches of different widths, yielding estimated diffusion and detachment rates similar to our previous measurements[17]. To estimate Cdc42-GTP rates, we used a GTP-locked form of Cdc42, Cdc42[Q61L]-mCherry[SW], expressed in addition to endogenous Cdc42 from the inducible *nmt* promoter, and performed photobleaching experiments early upon induction of expression on the sides of cells retaining the rod shape. This analysis confirmed a ~10-fold slower recovery of Cdc42-GTP, as previously observed[17]. Analysis using bleach zones of different widths at cell sides (Figure S3C) indicates that Cdc42-GTP detachment rates are reduced but non-zero, as also

previously suggested in *S. cerevisiae*[15]. We could not confirm a simultaneous reduction of Cdc42-GTP diffusion constant previously obtained by bleaching Cdc42 at cell tips[17]. Fitting FRAP profiles to extract two distinct rates carry significant uncertainties and orthogonal approaches will be required to measure more precisely Cdc42-GTP diffusion and detachment rates. In either case, the results show that Cdc42-GTP is less mobile than Cdc42-GDP by about one order of magnitude. In the simulations below, we use the values derived from Cdc42[Q61L]-mCherry[SW] FRAP analysis but also verified that results are robust when reducing Cdc42-GTP diffusion constant rather than detachment rate.

Simulations starting with half of all particles uniformly distributed on the surface, and reference parameter values (Tables S2, S3) result in spontaneous formation of a single, stably polarized state (Fig. 1E, Movie S1). The steady state concentration profiles of individual components (Fig. 1E) reproduce the peaks of CRIB-GFP and total Cdc42 observed experimentally in WT cells[17]. The steady state endocytosis and exocytosis profiles also match experimental measurements[29] (Figure S4). While the patch of active Cdc42 and GEFs is stable against membrane-driven flow away from the polarization site, sGAP is by contrast cleared away from the active region, similar to the cellular distribution of optoGAP, Rga4 and Rga6 along the cell sides[29,32–34].

Our model demonstrates how membrane-flow displacement of GAPs is likely an important regulator of pattern formation: removal of vesicle kinetics in the model results in an approximately uniform sGAP distribution and a significantly narrower activation region (Fig. 1F, left). Likewise, deletion of sGAP in the model results in loss of polarization through patch widening (Fig. 1F, middle), similar to Cdc42 patch and cell widening in cells lacking one or both of Rga4 and Rga6[32,33,36]. We note that the model without sGAP can form patches, depending on parameter values, and our reference parameter case is on the verge of such symmetry breaking. Indeed, a patch of Cdc42-GTP can also be generated upon moderate increase of $k_{hydro}$ in a simulation without sGAP, which can be considered to mimic overexpression of Rga3 in *S. pombe*, or reflect a situation more similar to *S. cerevisiae*, where there is no sGAP but several GAPs localizing at the Cdc42-GTP patch[3] (Fig. 1F, right).

## Reducing Cdc42 mobility leads to loss of polarity and viability

In previous work, we had shown that the C-terminal membrane-binding region of the mammalian GTPase Rit (ritC) homogeneously decorates the plasma membrane and exchanges rapidly with the cytosolic fraction. By contrast, tandem 2 or 3 ritC repeats lead to reduced detachment rate and progressive depletion from zones of polarized secretion[29]. Replacement of Cdc42 prenylation C-terminal CaaX box with 1ritC yields a viable allele, able to promote polarized growth, forming rod-shaped cells[17]. To experimentally probe the effect of lowering Cdc42 mobility, we thus constructed Cdc42-2ritC and Cdc42-3ritC variants, predicted to reduce Cdc42 mobility (Fig. 2A).

We first characterized further the Cdc42-1ritC variant. This and all other Cdc42 alleles described below were also internally tagged with mCherry, shown not to interfere with function[17]. We estimated the lateral diffusion and detachment rates of Cdc42-1ritC-GDP through FRAP experiments, which showed that Cdc42-1ritC has somewhat reduced mobility (largely due to reduction in detachment rate) relative to WT Cdc42, but is mobile enough to not be strongly affected by membrane flows (Fig. 2B)[29]. As previously shown, Cdc42-1ritC accumulated at cell poles (see Fig. 2C–F), similar to WT Cdc42[17]. These data establish the baseline against which we can compare Cdc42-2ritC and Cdc42-3ritC variants.

To study the effect of altering Cdc42, we constructed *cdc42-2ritC* and *cdc42-3ritC* alleles in diploid cells, keeping one WT copy of *cdc42*. Estimated lateral diffusion and detachment rates, based on FRAP experiments, indeed showed progressively slower mobilities (largely driven by slower detachment rates) of Cdc42-1ritC, −2ritC and 3ritC

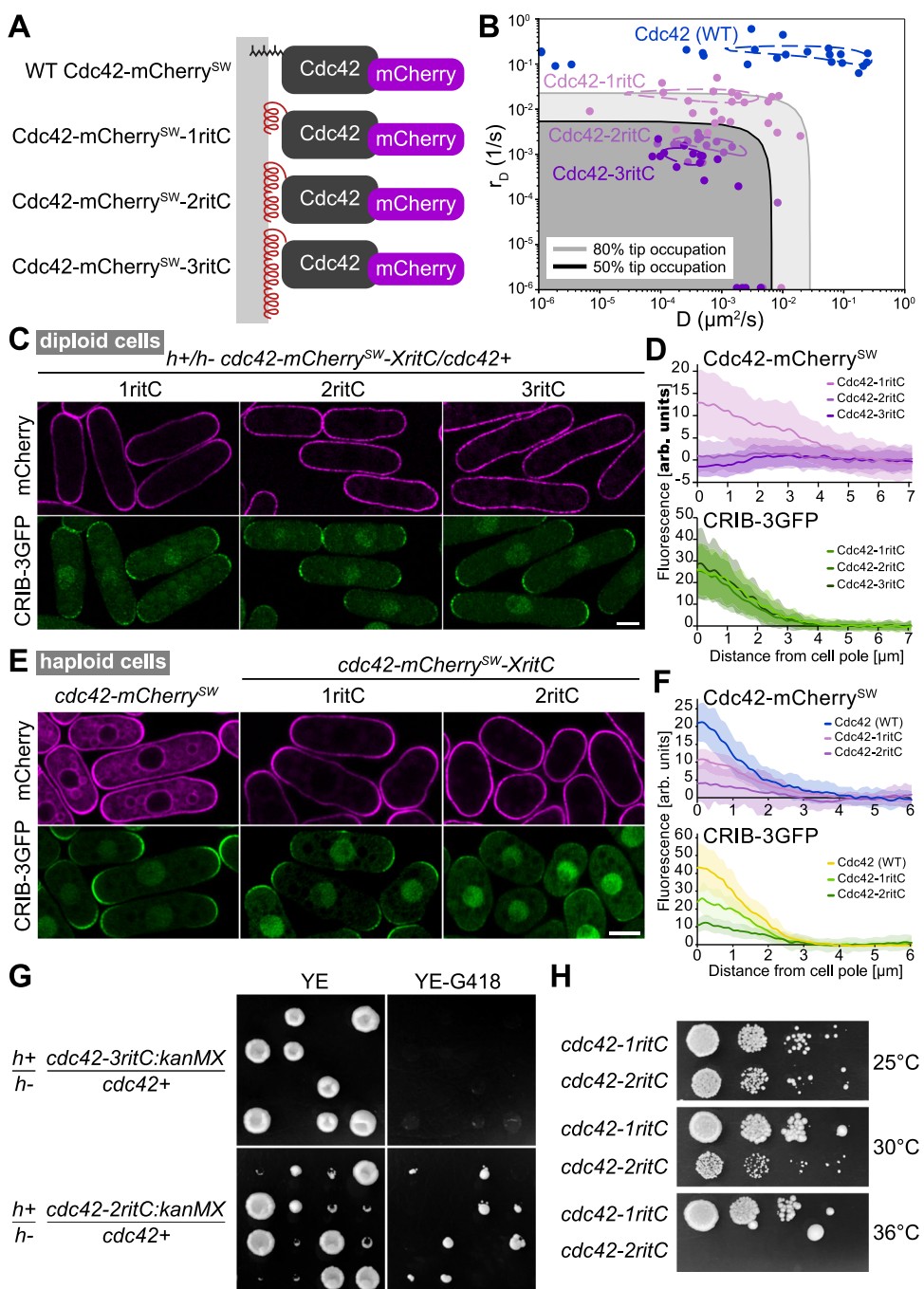

**Fig. 2 | Reduction of Cdc42 mobility at the plasma membrane compromises cell polarity and viability. A** Schematic figure of the constructed Cdc42 variants. The prenylation CAAX box is replaced by 1, 2 or 3 tandem copies of ritC (red), which targets Cdc42 peripherally to the plasma membrane (gray). Cdc42 is tagged with mCherry (purple) inserted in a poorly conserved region[17]. **B** D and $r_D$ (diffusion and membrane unbinding rates, respectively) of Cdc42-mCherry[SW] variants, derived from FRAP measurements. The measures were made at cell sides, which captures Cdc42-GDP dynamics. Points are best D and $r_D$ fits for individual cells. Dashed contour lines indicate 96% of the maximum $R^2$ value of the averaged $R^2$ plots for all fits. Gray shaded regions have predicted tip occupation below indicated levels. **C** Airyscan2 images of Cdc42-mCherry[SW] variants and CRIB-3GFP (marking Cdc42-GTP) in diploid cells also contain an untagged copy of WT *cdc42*. Representative image out of >100 imaged cells. **D** Average Cdc42 and CRIB cortical profiles from cell poles in diploid cells as in (**C**). Shaded areas show standard deviation. $n \geq 84$ half-tips for each strain. **E** Averaged airyscan2 images of Cdc42-mCherry[SW] variants and CRIB-3GFP (marking Cdc42-GTP) in haploid cells. Representative image out of >100 imaged cells. **F** Average Cdc42 and CRIB cortical profiles from cell poles in haploid cells as in (**E**). Shaded areas show standard deviation. $n \geq 24$ half-tips for each strain. **G** Tetrad dissections from indicated diploid strains. Growth on rich non-selective (YE) and *cdc42-mCherry[SW]-XritC*-selective (YE-G418) plates is shown. **H** Serial 10-fold dilutions of indicated strains on rich medium. Scale bars are 5 μm. For (**B**, **D**, **F**), source data are provided as a Source Data file.

(Fig. 2B), similar to the measured rates of GFP-tagged 1/2/3ritC[29]. In agreement with predictions due to membrane flows[29], reducing Cdc42 mobility led to perturbation of its accumulation at cell poles: in contrast to Cdc42-1ritC, Cdc42-2ritC and Cdc42-3ritC were not enriched at poles, with Cdc42-3ritC cells showing mild local depletion (Fig. 2C–D).

We note that all diploids exhibited similar distribution of CRIB (Fig. 2D), which marks the active Cdc42-GTP pool, and had similar width, indicating that the modified Cdc42 variants do not cause dominant perturbations to polarized cell growth. Thus, tandem dimerization of a membrane anchor can alter Cdc42 distribution,

lowering or preventing its accumulation at zones of polarized secretion.

As a test of our model, we performed simulations corresponding to diploid cells by doubling the volume and number of each particle type in the system compared to Fig. 1 (Figure S5). To simulate cells with *cdc42-ritC* alleles, half of Cdc42 had diffusion and dissociation constants corresponding to WT Cdc42 and half to Cdc42-1ritC, Cdc42-2ritC, or Cdc42-3ritC, respectively. All cases gave a single polarization site (Figure S5A) and the same (though somewhat stronger) trend of increasing Cdc42 depletion with decreasing Cdc42 mobility (Figures S5B–C, Fig. 2D). The model also predicted decreasing Cdc42 activation upon decreasing mobility, which we do not observe in experiments, suggesting that the activity of the WT Cdc42 variant dominates in cells.

To test the functionality of the low-mobility Cdc42 alleles, we sporulated the diploids and dissected tetrads. The *cdc42-3ritC/+* diploid gave rise to only two viable colonies, both lacking the *cdc42-3ritC*-associated selection marker (G418 resistance), indicating that *cdc42-3ritC* is inviable (Fig. 2G). The *cdc42-2ritC/+* diploid gave rise to four viable spores, two of which were G418-resistant and slow-growing (Fig. 2G). The *cdc42-2ritC* haploid cells were also temperature-sensitive, unable to grow at 36 °C (Fig. 2H). These cells exhibited aberrant, rounded or stubby cell shape, with increased and variable width, very little CRIB signal at the plasma membrane and almost no CRIB polarization (Fig. 2E–F). Similar to the case in diploid cells, Cdc42-2ritC was poorly enriched at cell poles (Fig. 2F). In summary, slow-down of Cdc42 mobility decreases Cdc42 activity and accumulation at cell poles, leading to loss of cell polarity and inability to sustain viability.

### Model reproduces experimental observations of Cdc42 mobility reduction

To understand how haploid Cdc42-2ritC cells polarize when Cdc42 itself is severely impacted by membrane flow while haploid Cdc42-3ritC are inviable, we repeated the simulations of Fig. 1 but with Cdc42-GDP diffusion and dissociation constants corresponding to those of Cdc42-1ritC, Cdc42-2ritC, and Cdc42-3ritC (Fig. 3A). The two slower Cdc42-ritC constructs (Cdc42-2ritC and Cdc42-3ritC) have lower diffusion and dissociation constants of their GDP-bound form than WT Cdc42-GTP (Figs. 1D, 2B, S3); thus, for the 2ritC and 3ritC cases we assumed that the corresponding mobility parameters of the GTP-bound form, in association or not with Scd1/2, are limited by the ritC anchor and set them the same as for the GDP-bound form.

Despite the opposing membrane flow, all three simulated Cdc42-ritC formed an active patch of enhanced Cdc42-GTP and depleted of sGAP, though very marginally for Cdc42-3ritC (Fig. 3A, Movies S2–S4). The profiles of total Cdc42 as well as Cdc42-GTP reproduce the experimental trends, showing decreasing concentrations with decreasing Cdc42 mobilities (Figs. 3B, 2F). Thus, GEF-mediated positive feedback in concert with flow-displacement of sGAP opposes the diluting effect of membrane flow on Cdc42, even for Cdc42 mobilities within the depletion region of Fig. 2B. However, reduction of Cdc42 mobility results in reduced overall Cdc42-GTP activation and a broadened patch, as shown in more detail in a systematic parameter scan of dissociation and diffusion constants (Fig. 3C).

### Reducing GAP activity restores viability to cells with slow Cdc42

The simulation results of Fig. 3 show that in regions corresponding to Cdc42-3ritC mobility, the number of Cdc42-GTP particles is very low. This observation leads to the hypothesis that a minimal level of Cdc42 activity, at a "survival" threshold above the Cdc42-3ritC parameters, is necessary for cell viability (Fig. 3C). Given the poor viability of Cdc42-2ritC, this survival threshold may overlap with the Cdc42-2ritC measured in diploid cells. We thus tested if a reduction in GAP activity in the model might bring Cdc42-3ritC above the survival threshold. Indeed, reducing the amount of sGAP particles by half or removing

sGAP completely enlarges the viability region (Figure S6) and brings Cdc42-3ritC simulated cells at or above the survival threshold (Fig. 4A). Strong reduction of $k_{hydro}$ to mimic *rga3* deletion also increases Cdc42-GTP levels and leads to narrowing of the patch, but the effect is weaker than that of reducing sGAP and insufficient to bring Cdc42 activity in *cdc42-3ritC* cells above the survival threshold (Fig. 4A).

To test experimentally if reducing Cdc42-GTP hydrolysis by GAPs restores viability to *cdc42-3ritC* cells, we constructed the *cdc42-3ritC* allele in a diploid strain that also carried heterozygous deletions of *rga3*, *rga4* and *rga6*. Sporulation of the strain would in principle permit recovery of haploid progeny carrying any combination of the *cdc42-3ritC* allele with *rga3Δ*, *rga4Δ* and/or *rga6Δ*. Interestingly, whereas sporulation of the *cdc42-3ritC/+* diploid did not produce G418-positive colonies (as also shown in the tetrad dissection, Fig. 2G), sporulation of the *cdc42-3ritC/+ rga3Δ/+ rga4Δ/+ rga6Δ/+* diploid produced G418-positive colonies, indicating recovery of viable *cdc42-3ritC* haploid cells (Fig. 4B). These cells were also mutant for *rga3Δ*, *rga4Δ* and *rga6Δ*, and we readily recovered haploid triple *rga3Δ rga4Δ rga6Δ* mutant cells that carried *cdc42-3ritC* as sole *cdc42* allele (Fig. 4C). These cells were very abnormally shaped, similar to the *rga3Δ rga4Δ rga6Δ* triple mutants carrying a WT *cdc42* allele[36], and had strong CRIB signal at the plasma membrane, showing that reduction of Cdc42-GTP hydrolysis by removal of Cdc42 GAPs restores Cdc42-3ritC activity. Thus, deletion of Cdc42 GAPs restores viability to a low-mobility Cdc42 allele.

## Discussion

Reaction-diffusion models have been successfully used to explain pattern formation on membrane surfaces. However, membranes are not inert surfaces, but may be actively remodeled by associated pattern-forming proteins, an aspect not generally captured in reaction-diffusion models. For instance, a local patch of Cdc42 GTPase activity promotes polarized secretion, inevitably leading to membrane flows away from the patch. Here, we developed a particle-based model to explicitly probe how membrane flows contribute to Cdc42 GTPase symmetry breaking. Our findings, supported by experimental validation, demonstrate that differential dynamics of fast Cdc42 and slow inhibitors provide a mechanism for pattern formation in the presence of membrane flow. In this mechanism, membrane flows enable protein dispersion that both constrains the membrane affinity of polarity factors and allows long-range displacement of inhibitors, implementing a transport-driven lateral inhibition mechanism as in classic activator-inhibitor pattern formation[43]. While prior studies showed that Cdc42-GTP relative low mobility traps it within its activation region[5,7], our study, as discussed below, reveals that Cdc42 diffusion coefficient on the membrane, and membrane dissociation rate, should nevertheless be fast enough for the average duration of Cdc42 within the active zone to be shorter than the characteristic time for membrane flow to displace it, in order to generate a zone with clearly defined boundaries.

The model demonstrates how membrane flows can be beneficial for polarization, by promoting the exclusion of a slow GAP. As a patch of Cdc42-GTP forms, local exocytosis promotes sGAP depletion, forming a zone permissive for Cdc42 activation. The slowness of GAP dynamics further helps to pin the location of the established activation zone. In the absence of membrane trafficking, sGAP is not depleted from the forming Cdc42-GTP patch and inactivates Cdc42. The simulations best reproduce experimental Cdc42-GTP patch formation in cells expressing optoGAP as sole Cdc42 GAP[29]. OptoGAP distributes homogeneously at the plasma membrane and turns off Cdc42 in the dark, but tightly binds the plasma membrane upon illumination, allowing its concomitant depletion from zones of polarized secretion and emergence of a Cdc42-GTP patch, similar to the emergence of the Cdc42-GTP patch in simulations. The endogenous GAPs Rga4 and Rga6 localize to cell sides and are excluded from zones of active secretion. We had previously shown that Rga4 localization is largely

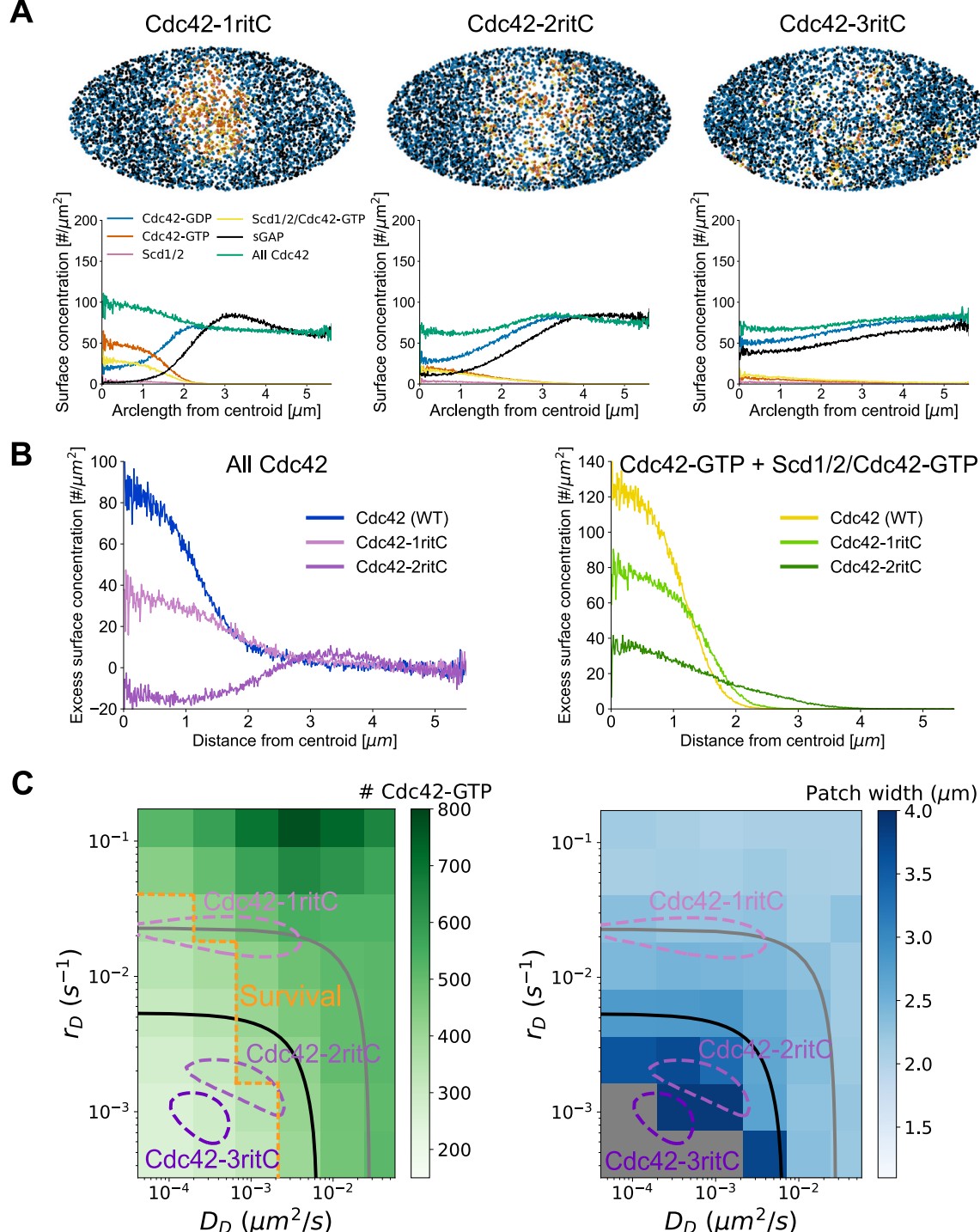

**Fig. 3 | Reduction of Cdc42 mobility in model leads to wider patches and lower Cdc42-GTP levels. A** Model results at steady state show that patch width increases as Cdc42 mobility is decreased to parameters for Cdc42-1ritC, Cdc42-2ritC, and Cdc42-3ritC as measured by FRAP. Snapshots taken after 8000 s. **B** Comparison of total Cdc42 levels and Cdc42-GTP levels (both normalized to side intensity) for Cdc42 (WT), Cdc42-1ritC, and Cdc42-2ritC as a function of distance from Cdc42-GTP patch centroid. **C** Parameter scan for membrane dissociation, $r_D$, and diffusion constant, $D_D$, for Cdc42-GDP, showing total Cdc42-GTP levels (left) and patch width (right). The Cdc42-1ritC, Cdc42-2ritC and Cdc42-3ritC contour lines and 50% and 80% depletion boundaries are drawn as in Fig. 2B. The postulated survival boundary is drawn at approximate Cdc42-2ritC model result; values with Cdc42-GTP levels below this line are postulated to lead to inviability. Gray regions in patch width plot indicate conditions where sGAP is not strongly affected by exo/endocytosis and no stable patch forms. Source data are provided as a Source Data file.

governed by membrane flow dynamics[29]. The localization of Rga6, which is slightly less depleted from cell poles than Rga4, and its dependence on polarized secretion[33], are also consistent with patterning by membrane flows, though this has not been directly tested. The contributions of these GAPs to Cdc42 patch formation are likely more complex as additional signals, such as lipid specificity or the cell cycle, also modulate their localization[44,45]. The role of Rga4/6 GAPs to corral the polarity site in fission yeast has strong similarities to that of septins, which define the bud neck in budding yeast, and whose displacement from the polarity site similarly requires polarized

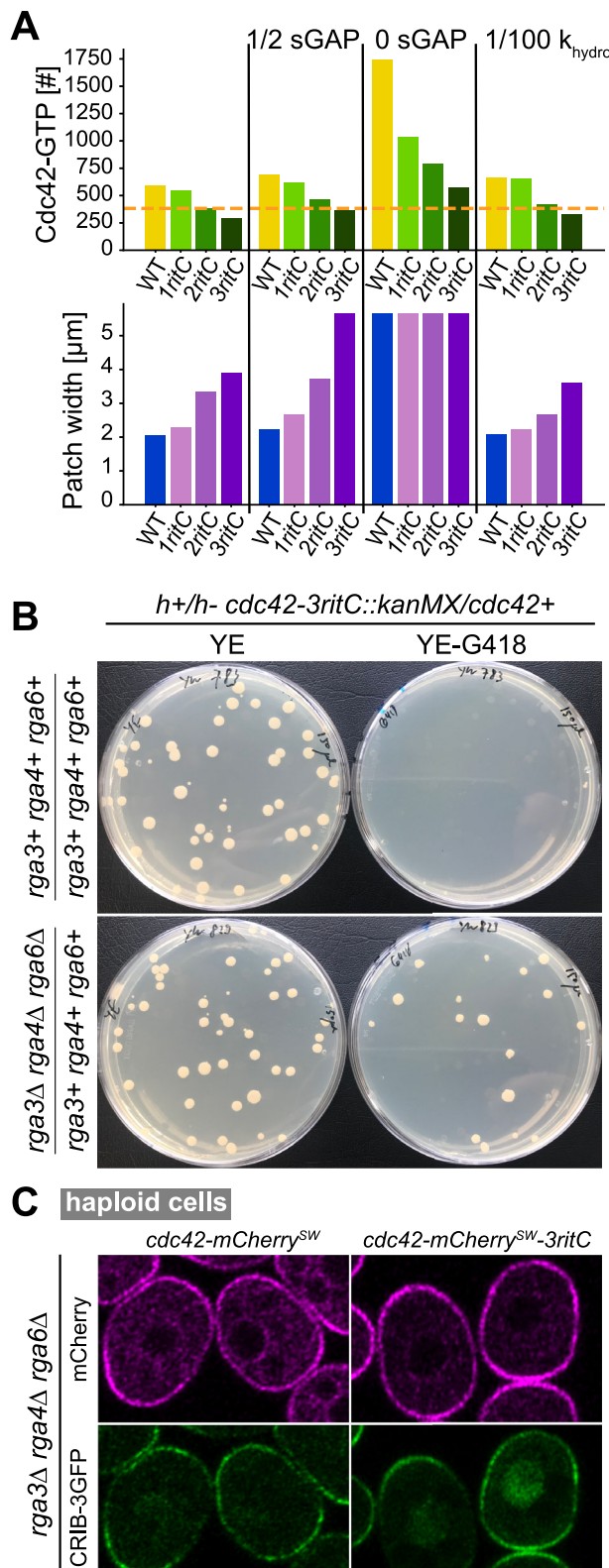

**Fig. 4 | Deletion of Cdc42 GAPs restores viability to strains containing low-mobility Cdc42. A** Model predictions for total amount of Cdc42-GTP and width of the polarity patch in reference conditions, upon reduction of sGAP and upon strong reduction of $k_{hydro}$ (mimicking *rga3Δ*). Source data are provided as a Source Data file. **B** Random spore analysis from heterozygous *cdc42-mCherry^SW^-XritC / cdc42+* diploids, which were otherwise WT (top) or contained heterozygous deletions for each of the three Cdc42 GAPs (bottom). Growth on rich non-selective (YE) and *cdc42-mCherry^SW^-XritC*-selective (YE-G418) plates is shown. **C** Airyscan2 images of Cdc42-mCherry^SW^ variants and CRIB-3GFP (marking Cdc42-GTP) in a haploid strain lacking all Cdc42 GAPs. Representative image out of >100 imaged cells. Scale bar is 5 μm.

membrane flows reinforces Cdc42-GTP patch formation in the natural in vivo context.

While membrane flows promote polarization through removal of negative factors, they also destabilize the polarity patch, which replenishes through rapid Cdc42 flux, in agreement with previous modeling effort[27,28]. We now show, both in silico and experimentally, that sufficiently fast dynamics (fast membrane unbinding rates and/or fast lateral diffusion) of Cdc42 is required for efficient polarization. The loss of polarization upon decreased Cdc42 mobility can be understood in the following way: Arrival of secretory vesicles at the patch causes local displacement and dilution of Cdc42, leading to widening and weakening of the patch. Consequently, secretory vesicle delivery is less polarized, reducing the net membrane flow. sGAP is thus less efficiently displaced, causing stronger overlap between sGAP and Cdc42-GTP and consequent Cdc42 inactivation. By contrast, fast dynamics allow Cdc42 to rapidly repopulate the zones of secretion, alleviating the effects of the membrane turnover. We did not delve into the influence of GEF turnover rates, but note that they were assumed to be fast in comparison to membrane flow; thus Scd1/2 recruitment by Cdc42-GTP can partly alleviate flow-induced displacement of low mobility Cdc42 (Fig. 3A, Cdc42-2ritC). We conclude that establishment and maintenance of a polarized patch requires its components (Cd42 and GEFs) to turn over faster than the surface on which it is formed.

In principle, a high Cdc42-GTP turnover at the membrane can result not only from fast Cdc42 protein mobility, but also from a high rate of GTP hydrolysis. Thus, an alternate mechanism for patch formation in the presence of flows that displace low mobility Cdc42-GTP, but in absence of sGAPs, may involve a higher level of $k_{hydro}$ (equivalent to Rga3 overexpression compared to our reference case, and provided that Cdc42-GDP mobility remains sufficiently high). In simulations, we indeed found that increasing $k_{hydro}$ improves polarization (see Fig. 1F), but only within a given range, as increased GTP hydrolysis also decreases the total amount of Cdc42-GTP at the membrane and eventually prevents patch formation altogether (Figure S7). As fission yeast cells have both slow (Rga4/6) and fast (Rga3) GAPs, this mechanism may also contribute and is implicitly included in the optimal $k_{hydro}$ value of our reference parameter.

The process of patch formation in our model is influenced by the localized reduction of sGAP, and therefore the results from our model cannot be entirely understood from the results of prior works that rely solely on non-linear positive feedback. This is shown directly from simulations where the elimination of the non-linear positive feedback (reduction of $\lambda_{STT}$ to 0 in the model) leads to a reduced patch size and strength but does not abolish polarization (Figure S8). This occurs for both WT and Cdc42-1ritC parameters. Therefore, the polarization in our model is in part driven by the localized reduction of the negative inhibitor on the surface; a mechanism very similar to one of the previously suggested, but experimentally unverified, classes of theoretical double-negative feedback models[6].

Additional flows may exist to aid cell polarization. For instance, Cdc42 is itself delivered to cell poles on secretory vesicles[17,38,47]. Thus, even though the concentration of Cdc42 in secretory vesicles was

secretion[46]. The position of endocytic sites around sites of secretion in polar-growing organisms may be similarly patterned, though the role of membrane flows in setting zones of endocytosis has not been directly investigated[30,40]. Fission yeast cells also express another Cdc42 GAP, Rga3, that colocalizes with the Cdc42-GTP patch[36]. Fine dissection of GAP localization and development of alleles that do not deplete from cell poles will be required to probe how much GAP depletion by

measured to be lower than that at the plasma membrane and dilute the polarity patch[27,38], secretory vesicles do not arrive "naked" with respect to Cdc42. While vesicle-based influx may not play a decisive role for the highly mobile WT Cdc42[28], it may be important for slower molecules. In previous work, we had shown that anchoring Cdc42 at the plasma membrane with a transmembrane anchor (Cdc42-TM), which cannot unbind the membrane but is delivered to cell poles on secretory vesicles, led to a viable allele, in contrast to the Cdc42-3ritC we present here[17]. Transmembrane proteins could be less perturbed by membrane flows in comparison to peripheral membrane proteins, due to interaction with structures that remain stationary with respect to lipid flow (accounted for by model parameter $\gamma > 1$). This suggests that membrane unbinding and rebinding is not the only mechanism that can efficiently repopulate Cdc42 at the poles.

We note that, with the Cdc42 rates we measured, the model predicts a somewhat stronger depletion of Cdc42-1/2ritC than we observe experimentally. Some of this discrepancy may stem experimentally from the selective pressure for viability of haploid cells. For instance, survivor *cdc42-2ritC* cells, at the edge of viability, may be selected to be in the high range of the Cdc42-2ritC mobility distribution. It is also likely that some aspects of Cdc42 polarization are not fully captured by the positive feedback and membrane flow simulations and that there are other positive inputs in the system, such as those played by microtubule-delivered factors[48,49]. There may also be effects of cell geometry. While most of our simulations used a sphere as a simple approximation of the hemispherical growing pole of fission yeast cells, without explicit growth simulation, we also simulated the WT Cdc42 and Cdc42-1ritC cases on the surface of a spherocylinder as shown in Figure S2. We find no large differences due to this change in morphology, demonstrating that our sphere simulations are a good approximation for polarization on a growing spherocylinder. However, we cannot exclude that other morphologies may yield unexpected consequences, as was previously observed with instability of the polarity patch at the tip of tube-like morphologies[50]. It would be interesting to investigate if displacement of negative regulators by membrane flows helps stabilize polarity in these morphologies, as was shown for mechanical feedback[51]. Simulations within an evolving shape to model growth would be informative in this respect.

Membrane flows are likely to play important roles in cell patterning across organisms[30]. For instance, whereas budding yeast cells do not exhibit Cdc42 GAPs depleted from the polarity patch, the formation of the septin ring, which forms a structural base of the bud neck and is proposed to associate with Cdc42 GAPs, involves central depletion by exocytosis[46]. A similar setup may be at play in pollen tubes, where the Rho GTPase inhibitor REN4 is displaced to the edges of pollen tips by the growth process[52]. Pollen tubes and many filamentous fungi grow much faster than *S. pombe*, which raises the question of whether the mechanism we propose here could maintain polarity on faster-flowing membranes. For instance, *Neurospora crassa* has an estimated 150-fold higher rate of membrane addition by exocytosis and 27-fold higher rate of removal by endocytosis than *S. pombe*[53]. Using these values and dimensions of the *N. crassa* hypha, we have calculated the boundary leading to 50% depletion from the hyphal tips using stationary profiles of endo- and exocytosis (equivalent to the region shown for *S. pombe* in Fig. 1D). While the boundary shifts to larger $D$ and $k_{off}$ values for *N. crassa*, compared to the *S. pombe* case, a dissociation rate similar to that of *S. pombe* Cdc42-GDP would provide sufficient mobility to withstand the membrane flow even in fast-flowing hyphae (Figure S9). Thus, the mechanism we propose may sustain, with only small adjustment of rates, a collectively "stable"−but rapidly turning over−polarity patch even on fast-flowing membranes. While cell polarity activators and inhibitors may vary across organisms, polarized secretion-induced membrane flows are likely to be intrinsic to polarization mechanisms across cell types.

## Methods

### Mathematical Modeling Methods and FRAP analysis

#### Computational model of Cdc42 polarization with explicit GAPs and membrane flow by exocytosis and endocytosis

**Spherical Simulation Domain and Parameters.** We simulated proteins as a system of particles on a surface representing the plasma membrane that migrate due to both reaction-diffusion and membrane delivery/removal by exocytosis/endocytosis. We focus on a single tip of the fission yeast cell, represented as a sphere with radius 1.8 µm. The particles are either restricted to the surface of this sphere or present in a cytoplasmic pool. Positions of particles in this cytoplasmic pool are not kept track of since we assume cytoplasmic diffusion is fast. The rate constants in the simulation were informed by FRAP experiments (this work), based on prior studies, or estimated as explained below (Tables S2, S3). Simulations were visualized with the Ovito software, version 3.0.0-dev334.

**Diffusion.** Particle diffusion on the spherical domain surface, with diffusion coefficient $D$, is implemented with Brownian dynamics as in[29]. Briefly, the magnitude of the displacement due to diffusion over time $\Delta t$ is calculated as $\sqrt{2D\Delta t}\xi(t)$ independently in both directions in the plane tangent to the sphere at the original particle location, and $\xi(t)$ is a random Gaussian distributed variable with zero mean and unit variance. After calculating the displacement in the tangent plane, the positions of each particle are then projected down to the sphere surface while keeping the distance conserved[54].

**Particle reactions.** Reactions of diffusing particle species in the simulation include membrane binding and unbinding, unimolecular reactions (indicated with rate constants symbols $k,r$ in Fig. 1A), and bimolecular reactions on the membrane (indicated by $\lambda$ in Fig. 1A). When particles on the sphere undergo an unbinding event, they are removed from the sphere surface and placed in an internal pool representing the cytoplasm. Particles in this internal pool are placed back on the surface anywhere with uniform probability according to the association rate constants. Bimolecular reactions in the model include discrete sGAP hydrolysis, Scd1/2/Cdc42-GTP trimer complex formation, and promotion of Cdc42 to its active form by either Scd1/2 or Scd1/2/Cdc42-GTP. These reactions depend on the existence of two particles, and most are indicated by reactions that have a $\lambda$ variable indicating their rate (Fig. 1A). Bimolecular reactions with rate $\lambda$ are implemented using the Doi method where two species within a microscopic distance, $\rho$, react with rate $\lambda$[41,55-57]. Using this method, $\rho$ and $\lambda$ together define the macroscopic rate constant. For Scd1/Scd2/Cdc42-GTP formation via cytoplasmic Scd1/2 binding to membrane-bound Cdc42-GTP, we calculate the rate by the rate constant $k_{ST}$ instead of the Doi method (since the positions of cytoplasmic particles are not defined in our model). When particles form a complex (reaction between Scd1/2 and Cdc42-GTP particles, both bound to the membrane), the complex is placed at the midpoint between the prior positions of the two reactants. For the reaction between cytoplasmic Scd1/2 and membrane-bound Cdc42-GTP we place the resulting complex at the position the Cdc42-GTP had since it is the only particle on the membrane for this reaction. Dissociation from the membrane places the molecules in the cytoplasmic pool. When complexes separate, the separated particles are placed at distance $\sigma$, slightly larger than $\rho$. Hydrolysis and activation reactions change the type of Cdc42 particle with the sGAP and Scd1/Scd2 complex positions unchanged. The reactions that occur in our model are given in Table S1.

**Assumptions of particle reaction scheme.** (1) The rate constant of hydrolysis $k_{hydro}$ represents the combined effect of spontaneous hydrolysis and hydrolysis by the GAP protein Rga3, which is known to colocalize with Cdc42-GTP to the tips of *S. pombe* cells[36]. We assume that $k_{hydro}$ is uniform in space. (2) Hydrolysis by slow GAP (sGAP)

represents hydrolysis by Rga4/6 which localizes to the sides of *S. pombe* cells in part due to membrane flows[29]. For simplicity, and without loss of generality, we assume sGAP has low mobility with a diffusion coefficient of $10^{-3}$ µm²/s and no membrane unbinding, causing it to be influenced by membrane flows and redistribute away from patches of active Cdc42 (locations of exo/endocytosis). Initially half of the sGAP is in the cytoplasm, binding irreversibly to the plasma membrane with a rate constant of 1/30 s⁻¹. (3) Reactions involving Scd1/2 represent a minimal version of the positive feedback required for symmetry breaking[19] that approximates the sum of separate contributions of Scd1, Gef1, and Ras1[10,58,59]. (4) The membrane binding rate of Cdc42-GDP, $k_D$, was set dependent on the dissociation rates $r_T, r_D$ of Cdc42 GTP/GDP so that approximately 3/5 of Cdc42 is on the membrane. We determined the functional form of this dependence from the steady state balance equation $k_D N_{cyto} = N_T r_T + N_D r_D$, where $N_{cyto}, N_T, N_D$ are the number of Cdc42 in the cytoplasm, and bound to the membrane with GTP or GDP, respectively. Additionally, we fix $N_D = 2\beta N_{cyto}$ and $N_{cyto}/(N_{cyto} + N_D + N_T) = 1/(1+3\beta)$, where for our simulations $\beta = 0.513$ in order to have a ratio of 3/5 of the total Cdc42 on the membrane. Solving these three equations leads to the form shown in Table S2: $k_D = \beta(r_T + 2r_D)$. Indeed, for the reference WT-Cdc42 simulations $N_D \approx 2200, N_{cyto} \approx 2220, N_T \approx 380$, and $N_{scd12T}^{WT} \approx 200$.

**Membrane flows.** Particle movement by exocytosis and endocytosis was implemented as in prior work[29]. Briefly, we assume that lipid traffic exocytosis and endocytosis near cell tips create a flow of lipids and of all associated proteins included in the model along the plasma membrane. We do not incorporate a full hydrodynamic model, assuming the instantaneous displacement of proteins around the site of the vesicle event[27,60]. This assumption is supported by the appearance of a flat plasma membrane in electron micrographs of fission yeast cells at cell sides and tips, except at sites of vesicle traffic or around eisosomes[61–63], similar to *S. cerevisiae*[62,64]. This observation indicates the presence of a membrane tension, which, combined with turgor pressure, can flatten the vesicle membrane delivered by exocytosis[65,66], causing lateral flow. Similarly, flow of lipids towards endocytic vesicles must occur over a sufficiently wide area, preventing membrane strains larger than a few % that would otherwise lead to formation of membrane pores[67].

**Our model incorporates three coarse-grained hydrodynamic parameters.** (1) Parameter $\alpha$ describes the mobile fraction of lipids in the plasma membrane; where $\alpha = 1$ indicates that the entire membrane is able to flow. We used $\alpha = 0.5$. (2) Parameter $\gamma$ measures the ratio of lipid to membrane-bound protein velocity, which could be larger than unity due to interactions of proteins with structures that remain stationary or less mobile with respect to lipid flow, such as in the cell wall or in the cytoplasm. We used $\gamma = 1$. (3) Parameter $R_{cutoff}$ is a cutoff distance to the flow induced by individual exocytosis and endocytosis events, accounting for slow equilibration of membrane tension over long distances[68]. We used $R_{cutoff} = 2$ µm. Protein depletion by membrane flow is robust with respect to the precise values of $\alpha$, $\gamma$, and $R_{cutoff}$[29].

Lipids on the surface at arc length distance $s$ from the center of an exocytosis event are assumed to be pushed away radially by arc length distance $\Delta r_{exo/endo}(s)$ as given below where the exocytosis displacement is the top sign and endocytosis is the bottom sign where a plus and minus are present[29].

$$\Delta r_{exo/endo}(s) = \frac{1}{\gamma}\left\{ \pm R' \text{acos}\left[1 - \left(\frac{R}{R'}\right)^2 \left[1 - \cos\left(\frac{s}{R}\right)\right] \mp \frac{A_{exo/endo}}{2\pi R'^2 \alpha}\right] \mp s \right\},$$

$$R' = \sqrt{R^2 \pm A_{exo/endo}/4\pi}$$

(1)

This displacement is calculated by assuming that the difference between the spherical cap surface area corresponding to mobile lipids measured at the final and initial arc length positions of the particle from the center of exocytosis or endocytosis equals the surface area of the exocytotic or endocytic vesicle, $A_{exo}$ or $A_{endo}$, respectively. The displacement is only applied to particles up to an upper arc length distance, $s < R_{cutoff}$. For an endocytosis event, particles within the cutoff distance $R_{cutoff}$ are moved radially toward the endocytic point (or exactly at the endocytic point if they happen to be sufficiently close to it). We thus assumed that the particles of our model do not get internalized by endocytosis.

**Exo/endocytosis distributions.** We assume that both exocytosis and endocytosis events occur with constant overall rates, directed towards sites of membrane-bound Cdc42-GTP. Specifically, that each Cdc42-GTP enhances the probability of an exo/endocytosis event occurring near it according to a Gaussian distribution as function of arc-length distance to the Cdc42-GTP particle, with standard deviation $w_{exo}$ or $w_{endo}$, respectively. The probability distribution of an event occurring anywhere on the simulation domain is the sum of Gaussian distributions centered on each Cdc42-GTP particle position (Fig. 1B). We tuned the values of $w_{exo}$ or $w_{endo}$ to reproduce the experimentally-measured exocytosis and endocytosis profiles around the cell tip as determined from Exo70-GFP and Fim1-mCherry[29], given the experimentally-measured profile of Cdc42-GTP measured by CRIB-GFP[17]. This implies $w_{exo/endo} = \sqrt{\sigma_{exo/endo}^2 - \sigma_{Cdc42-GTP}^2}$, where $\sigma_{exo/endo}^2$ is the variance of the experimental exocytosis/endocytosis profile and $\sigma_{Cdc42-GTP}^2$ is the variance of the experimental CRIB profile as function of arc-length distance from the cell tip. This tuning gives $w_{endo} = 1.51$ µm. Since the experimental exocytosis profile is slightly narrower than the Cdc42-GTP profile, $w_{exo}$ is set to a small value (0.01 µm). At reference parameters, the model generates exo/endocytosis distributions that are in good agreement to experimentally measured profiles (Figure S4).

**Time Step.** A fixed timestep $\Delta t = 10^{-3}$ s was used for reactions and Brownian diffusion dynamics, except for exocytosis/endocytosis and unimolecular membrane binding reactions that were based on a varying timestep determined by the Gillespie algorithm, independent of this discrete timestep (where the time until the next reaction is a continuous variable and the reaction that occurs is chosen randomly according to the sum of all transition rates that can occur through this method[69]). An upper bound $\Delta t$ of $7.35 \times 10^{-3}$ s was calculated from the root mean squared displacement equation for diffusion on 2D surfaces, $\rho \approx \sqrt{2D\Delta t}$, where $D$ is the largest diffusion constant used in the simulation (0.17 µm²/s). We truncated this upper bound to determine a timestep of $10^{-3}$ s sufficient for our reaction parameters. Additionally, for the largest $\lambda$ we consider a bimolecular reaction occurs on the first step after entering the reaction radius, $\rho$, slightly less than 1% of the time. The timesteps determined from the Gillespie algorithm occur in between discrete timesteps. We use the Gillespie algorithm for exo/endocytosis events that occur less frequently. Since membrane binding reaction rates do not depend on spatial position, and could also occur rarely (such as for Cdc42-3ritC), we also used a Gillespie algorithm for these reactions. Diffusion for every particle in the system occurs to advance the system up to discrete timesteps and to times associated with exo/endocytosis. To handle the split between Gillespie and discrete timestep, each particle has two times associated with it; $t_{react}$, which is the most recent time the particle was allowed to react up to, and $t_{pos}$ which gives the most recent time the particle was diffused up to. Specifically, the probability to react is determined by $P_{react} = 1 - \exp\left[-r(n\Delta t - t_{react})\right]$, where $n$ is the number of discrete time steps and $r$ is the reaction rate. Particles are diffused over the time period of $t - t_{pos}$, where $t$ is the current simulation time. Upon

membrane binding of a specific particle, both $t_{react}$ and $t_{pos}$ are set equal to the current simulation time.

**Simulations with varying Cdc42 mobility.** In simulations with lowered Cdc42-GDP diffusion coefficient and membrane dissociation rate constant, which includes simulations of Cdc42-XritC constructs, we also lowered the corresponding values of Cdc42-GTP and Scd1/2/Cdc42-GTP so that Cdc42-GDP would always have greater than or equal mobility compared to Cdc42-GTP. Specifically, the three following equations hold throughout all simulations $D_T = \min\{D_D, D_{T,WT}\}$, $r_T = \min\{r_D, r_{T,WT}\}$, and $D_{ST} = \min\{D_T, D_{ST,WT}\}$ where subscript WT indicate WT values (Table S2). The membrane binding rate of Cdc42-GDP, $k_D$, was set so that approximately 3/5 of Cdc42 is on the membrane as described in *Assumptions of particle reaction scheme*. We note that this calculation of $k_D$ assumes that the binding affinity of Cdc42-XritC constructs for the plasma membrane remains approximately the same as for WT cells. We checked that our results do not change when $k_D$ remains constant (Figure S10).

**Diploid simulations.** For diploid simulations, we doubled the volume and the number of each species in the system and assigned half of these the WT parameters for Cdc42 unbinding and diffusion while the other half was assigned dissociation and diffusion constants for a different reference case of either Cdc42-WT, Cdc42-1ritC, Cdc42-2ritC, or Cdc42-3ritC.

**Spherocylinder simulations.** To explore the assumptions of our sphere model, we performed simulations that included volume and surface expansion, implementing particle-based reactions along a growing cell tip. Simulations with local growth and membrane flow as a result of local Cdc42 activity are challenging[50,51] so we simplified the problem by assuming a fixed geometry and working in the reference frame where the cell tip is at rest. Rather than implementing discrete exo/endocytosis events, in these simulations each particle is displaced radially along the spherical cap or cylindrical segment according to an azimuthally symmetric velocity that accounts for cell growth and net membrane area added or removed. Assuming the center the Cdc42-GTP patch is the tip of the spherocylinder, this velocity flow profile, shown in Figure S2A, was determined by: (1) Freezing Cdc42-GTP particles in a patch with width equal to the average patch from steady state WT/1ritC simulation on the sphere surface as in Figs. 1E and 3A. (2) Running the simulation with discrete endocytosis and exocytosis events generated by the stationary Cdc42-GTP and calculating the induced flow on probe ghost particles placed uniformly on the sphere. The ghost particles were used to accumulate the average flow and were not displaced during these simulations.

For the spherocylinder simulations (Figures S2B–D) we began with the same surface density and number of particles as our sphere simulations. We initialized the cylinder length to be 3.0 µm with the spherical radius remaining at 1.8 µm as in all other simulations. For each timestep, we determined the displacement on the spherocylinder surface due to the flow velocity (Figure S2A) and diffusion. We then displaced the particle on the surface it begins on (e.g., hemisphere or cylinder), and if the particle crossed the boundary between two surfaces, we advanced the particle up to this boundary and continued the displacement on the other surface. Reflecting boundary conditions were applied at the free edge of the cylindrical segment. We performed simulations where the reflecting boundary was moved at a rate equal to the rate of tip growth, or else kept stationary. Since cell growth is relatively slow compared to the time of steady patch formation, we kept the number of particles fixed throughout the simulation. To compare the spherocylinder geometry results, both with or without explicit growth, to results using spherical geometry, we implemented the same algorithm for a sphere, assuming the flow originates at a fixed location on the sphere (Figure S2C–D).

**Quantification of particle concentration profiles.** Particle concentration profiles (e.g., Fig. 1E) were measured by calculating the arclength distance from the centroid associated with the largest cluster of Cdc42-GTP. Cdc42-GTP particles were placed in the same cluster if their arclength distance was smaller than 0.3 µm. The instantaneous centroid was calculated by averaging the positions of Cdc42-GTP in this largest cluster, and then projecting back to the sphere surface. The centroid used for the profiles was calculated from an average over five frames of the instantaneous largest centroid. The arclength of the particles on the sphere was measured from this centroid and then placed into histograms for each particle type. The values in this histogram were divided by the area of the strip on the sphere surface to determine the surface density for each particle. Profiles were averaged over at least 500 s.

**Quantification of polarization patch width.** Patch widths were calculated based on the sGAP profile. Specifically, by determining the arclength from the Cdc42-GTP centroid value at which the sGAP concentration first drops below 90% of the concentration sGAP has far from the patch centroid. The far field concentration was determined by averaging the sGAP concentration over 5.05 µm to 5.55 µm arclength from the patch centroid. The arclength of the first drop below 90% was determined by starting from an arclength of 5.05 µm and moving towards 0 µm, averaging concentration over a moving window of 0.1 µm. Cases where the concentration of sGAP did not drop below 90% of the far field value were marked as unpolarized (Fig. 3C, S6B) or the patch width was set equal to half the circumference of the sphere, 5.65 µm (Fig. 4A).

## Determination of membrane diffusion and dissociation rates by FRAP

Effective diffusion coefficients $D$ and membrane dissociation constants $k_{off}$ for WT and Cdc42 mutants were estimated by fitting the experimental recovery profiles to a 1D model as described in prior work[29] and summarized below, using the code available at https://github.com/davidmrutkowski/1DReflectingDiffusion.

*FRAP of cell sides.* Experimental fluorescence intensity profiles were acquired after photobleaching of a rectangular region and imaging on the top surface of *S. pombe* cells (Figure S3A). Since Cdc42 is primarily inactive at cell sides[17], we assume the fit for $D$ and $k_{off}$ corresponds $D_D$ and $r_D$ for Cdc42-GDP. As detailed in the experimental methods section, we proceeded in a similar manner to derive $D_T$ and $r_T$ for Cdc42-GTP from the recovery profiles of a GTP-locked allele of Cdc42, Cdc42[Q61L]. This fit for two parameters relies on $D$ and $k_{off}$ contributing differently to the recovery profile, with diffusion providing smoothing of profile over time and association/dissociation contributing to uniform recovery (in the limit of fast cytoplasmic diffusion). The values of these two parameters were further narrowed down by performing FRAP of different bleach widths, which have varying relative recovery contributions of diffusion and association/dissociation.

We corrected for photobleaching by multiplying all intensities by $e^{kt}$ after background subtraction, where $k$ is the exponential decay constant of the average intensity of a neighboring non-bleached cell. From these photobleach-corrected images we determined a 1D intensity profile $I(x,t)$ (where the x-dimension corresponds to the long axis of the cell) by defining a rectangular region of interest along the long axis of a bleached cell with width ~2 µm, excluding the curved region close to the cell tips, and calculated the average intensity over time along this line at points spaced 0.0516 µm (1 pixel) apart using ImageJ's getProfile function.

Each FRAP fit to $I(x,t)$ results to individual best fit points in diffusion and dissociation constant space (Figs. 2B, S3C). The dashed contour lines in Figs. 2B, S3C were determined by averaging together the normalized $R^2$ measure of fits for each FRAP curve and finding the contour line that traces out 96% of the maximum. The maximum value

of this average normalized $R^2$ was taken as the best fit value for as the diffusion and dissociation constants.

*Model with diffusion and association/dissociation used for fitting.* We modeled the evolution of the concentration profile of a membrane-associated protein, $c_m(x,t)$, along a 1D line with reflecting boundaries at positions $x_a$, $x_b$, defining the finite size of a membrane domain. We assumed $c_m(x,t)$ obeys the following equations, which include both the effects of diffusion and membrane binding/unbinding:

$$\frac{\partial c_m(x,t)}{\partial t} = D\frac{\partial^2 c_m(x,t)}{\partial x^2} - k_{off}c_m(x,t) + k_{on}C(t) \quad (2)$$

$$C(t) = C_{tot} - \int_{x_a}^{x_b} c_m(x,t)dx, \quad (3)$$

where $c_m$ is concentration on the membrane, $C$ is the number of proteins in the cytoplasm, $C_{tot}$ is the total number of proteins, and $k_{off}$, $k_{on}$ are the membrane dissociation and association rate constants. We do not write a separate diffusion equation for the cytoplasm, assuming that cytoplasmic diffusion is sufficiently fast. We also specifically consider the limit where most of the proteins are on the membrane: $C(t) \ll C_{tot}$, (equivalently, the limit of sufficiently large $k_{on}$). The Green's function describing the probability for a single protein being at position $x$ at time $t$, given that it was at position $x_0$ at time $t = 0$ is

$$p(x,t|x_0,0) = e^{-k_{off}t}p_D(x,t) + \frac{1-e^{-k_{off}t}}{x_b - x_a}, \quad (4)$$

where $p_D(x,t)$ accounts for the probability of the particle position due to diffusion alone:

$$p_D(x,t|x_0,0) = \frac{1}{x_b - x_a} + \frac{2}{x_b - x_a}\sum_{n=1}^{\infty} e^{-Dt\lambda_n^2}\cos(\lambda_n(x - x_a))$$
$$\cos(\lambda_n(x_0 - x_a)), \text{ where } \lambda_n = \frac{n\pi}{x_b - x_a} \quad (5)$$

*Fitting to experimental I(x,t) profiles.* Assuming $c_m(x,t)$ is proportional to the experimental $I(x,t)$, we solve for $c_m(x,t)$ given an initial condition $c_m(x,0)$ that is proportional to the intensity profile imaged immediately after the photobleaching, $I(x,0)$. We then fit the model to both $x$ and $t$ after photobleaching simultaneously, using SciPy's optimization curve fitting to get a single best fit value for $D$ and $k_{off}$ as described below. The sum in Eq. (3) was carried up to $n = 500$. The intensity profile is divided into three regions: left, middle, and right. The intensity in the middle region, defined to be between $x_d$ and $x_e$ where $x_a \leq x_d \leq x_e \leq x_b$, is determined by the intensity profile $I_i^0$ immediately after photobleaching, evaluated at pixels $i$ of the projected profile, corresponding to positions $x_i$. The middle region is therefore treated as a sum of $n = \Delta x/(x_e - x_d)$ separate integrals, where $\Delta x$ is the pixel size. The lengths of the left and right flanking regions are both equal to $l_F$, with $l_F = x_d - x_a = x_b - x_e$. We average the intensity of the pre-bleached cell over the middle region and set this as the initial intensity of the two flanking regions, $I_F^0$. The value of $l_F$ determines the uniform intensity at long times, $I_\infty$, according to mass conservation:

$$(2l_F + x_e - x_d)I_\infty = \sum_{i=0}^{n} I_i^0 \Delta x + 2l_F I_F^0. \quad (6)$$

The long time intensity $I_\infty$, or equivalently $l_F$, is determined via the fitting procedure. When the $l_F$ value drifted towards 0 or large numbers, it was restricted to be in the range 2-10 μm. The equation describing the concentration or intensity as a function of position $x_i$ corresponding to pixel $i$, and time is found by analytical integration, $I_{model}(x_i,t) = \int_{x_a}^{x_b} I(x',0)p(x_i,t|x',0)dx'^{29}$ with the initial intensity $I(x,0)$ at $x_i < x < x_{i+1}$ given by $0.5(I_i^0 + I_{i+1}^0)$. An example of a fit over time to

the experimental FRAP data for one cell with Cdc42-WT is shown in Figure S3B.

## Experimental methods

**Media and growth conditions.** For microscopy experiments, cells were grown to exponential phase at 25 °C in Edinburgh minimal medium (EMM) supplemented with amino acids as required, except diploids which were grown in complete YE medium to repress sporulation. For Cdc42$^{Q61L}$-mCherry$^{SW}$ expression, the strain was grown to log phase in EMM-AU + thiamine, washed 3x in medium without thiamine, diluted in EMM-AU without thiamine and grown to log phase for a further 20–22 h at 25 °C. Tetrad dissections and dilution series were grown on YE plates at 25 °C, unless otherwise indicated. Random spore analysis was done by digestion of sporulated diploids with glusulase, growth on YE plates at 25 °C and replication on YE plates containing appropriate antibiotics.

**Strain construction.** Strain genotype is given in Table S4. Integrative plasmids carrying the *cdc42-3ritC* and *cdc42-2ritC* alleles were constructed as follows. Plasmid pSM1442 (pFA6a-3'UTR(*cdc42*)-cdc42-mCherry$^{SW}$–1ritC-Term-kanMX[17]) was digested with NheI-XmaI to remove the 1ritC fragment. The 3ritC fragment, amplified with primers osm7870 and osm7871 from pAV601, and the 2ritC fragment, amplified with primers osm7870 and osm7872 from pAV602[29], were cloned by InFusion cloning, yielding plasmids pSM2934 (pFA6a-3'UTR(*cdc42*)-cdc42-mCherry$^{SW}$–3ritC-Term-kanMX) and pSM2935 (pFA6a-3'UTR(*cdc42*)-cdc42-mCherry$^{SW}$–2ritC-Term-kanMX), respectively. Sequences of these plasmids are provided as Supplementary Data 1 and Supplementary Data 2. These plasmids were linearized with SalI and integrated in diploid strains constructed using the trans-complementation of *ade6-M210* and *ade6-M216* alleles. In each case, the diploids were selected on EMM, and propagated on YE plates, before transformation and selection on YE-G418 plates.

To estimate the mobility of Cdc42-GTP, we used a strain in which the native cdc42 is tagged with sfGFP (*cdc42-sfGFP$^{SW}$*) and GTP-locked Cdc42$^{Q61L}$-mCherry$^{SW}$ is expressed from an episomal plasmid under *nmt41* promoter (YSM2615[17]).

The *cdc42-2ritC* and *cdc42-3ritC* alleles in otherwise WT diploid strains were obtained by transformation of linearized pSM2934 and pSM2935 in a strain obtained by crossing haploids *h+ ura4-D18 ade6-M210* (YSM4072) and *h- ura4+ ade6-M216* (YSM4073), yielding strains YSM4074 and YSM4075, respectively (Fig. 2G). The same alleles in a strain carrying the CRIB marker were obtained by transformation in a diploid obtained by crossing haploid strains *h- ura4-D18 ade6-M210 leu1-32:pshk1:CRIB-3GFP:ura4 + :leu1 +* (YSM4076) with *h+ ura4-D18 ade6-M216 leu1-32* (YSM1182), yielding strains YSM4077 and YSM4078, respectively. The *cdc42-3ritC* allele in a strain carrying deletion of genes encoding Cdc42 GAPs was obtained by transformation of linearized pSM2934 in a diploid obtained by crossing *h+ ura4- ade6-M216 leu1- rga4Δ::natMX rga3Δ::hphMX* (YSM4079) and *h- ura4-D18 ade6-M210 leu1-32:pshk1:CRIB-3GFP:ura4 + :leu1+ rga6Δ::bleMX* (YSM4080), yielding strain YSM4081.

The diploid strain carrying the *cdc42-1ritC* allele was obtained by crossing a derivative of the *cdc42-1ritC* strain described in[17], *h+ ura4-D18 ade6-M216 leu1-32 cdc42-mCherrySW-1ritC:kanMX* (YSM4082), with YSM4076, yielding strain YSM4083.

Haploid *cdc42-2ritC* strains were obtained through sporulation and tetrad dissection of diploids: YSM4084 from sporulation of YSM4075 (Fig. 2H); YSM4085 from sporulation of YSM4078 (Fig. 2E); and a haploid *cdc42-3ritC* strain *rga3Δ rga4Δ rga6Δ* deletion (YSM4086) from sporulation of YSM4081. As this strain did not contain the CRIB transgene, this was re-integrated yielding strain YSM4087 (*rga3Δ rga4Δ rga6Δ*) (Fig. 3C). Diagnostic PCRs distinguishing WT from *cdc42-mCherry-3ritC* alleles were used to further to verify the Cdc42 allele present in this strain.

**Microscopy.** Cells were imaged with Plan-Apochromat 63×/1.40 oil differential interference contrast objective on an LSM980 system equipped with an airyscan2 detector and acquired by the ZEN Blue software (ZEN 3.3; Zeiss). Imaging was set in SR mode with frame bidirectional scanning. Laser power was kept at <0.6%, with pixel time around 13.2 μs, and frame size of 634×634. All other settings were optimized as recommended by the ZEN Blue software. Images in Figs. 2E and 4C were acquired with 4x averaging to obtain smoother distributions of CRIB and Cdc42. FRAP at cell middle was performed on cells placed on an EMM-2% agarose pad by finding the focal plane closest to the coverslip and defining a rectangular region of interest of defined dimension (1 to 6 μm wide) in the middle of the visible signal (see Figure S2A). FRAP of Cdc42$^{Q61L}$-mCherry$^{SW}$ was performed on cells that retained the rod shape to ensure that polarity was not yet grossly disrupted by the uniform Cdc42 activity at the cell periphery. We had shown in previous work that Cdc42$^{Q61L}$-mCherry$^{SW}$ does not alter the dynamics of native Cdc42-sfGFP$^{SW}$ ref. 17. Two snapshots were taken before bleaching, which was done using a single pass of the bleach zone at 100% laser power. Recovery was imaged for up to 5 min at intervals adapted to the speed of recovery of each protein. All Airyscan images were processed in ZEN Blue software.

**Image analysis.** Profiles of Cdc42 and CRIB at the cell periphery were acquired in FIJI (using ImageJ 1.54 f) from 6 μm-long (7μm-long for diploid cells), 5 pixel-wide segmented lines starting at the cell pole with higher CRIB signal (2 profiles per cell) and averaged. To estimate the amount of fluorescence signal captured in the cortical profiles, we assumed that any CRIB-GFP signal in cortical profiles at the middle of interphase WT cells represents cytosolic signal. We found that this corresponds to about half of the background cytosolic fluorescence measured in a small region devoid of vacuoles. We thus removed cytosolic signal from cortical profiles by subtracting 0.5 x the cytosolic fluorescence measured in a small region devoid of vacuoles (using Excel 16.88). The CRIB profiles plotted in Figs. 2D and 2F are averages (≥24 profiles for averaged images in 2 F; ≥84 profiles for non-averaged images in 2D). To compare the tip to side distribution of Cdc42, we subtracted the average value at distance 5–6 μm from the cell pole from the Cdc42 profiles. In analysis of microscopy images, no data were excluded, except out-of-focus images or timelapse with important drift during acquisition. No statistical method was used to predetermine sample size. The experiments were not randomized.

### Reporting summary

Further information on research design is available in the Nature Portfolio Reporting Summary linked to this article.

## Data availability

Source data are provided with this paper. A minimum dataset comprising the microscopy images shown and quantified in Figs. 2C–E and 4C is available on Figshare [https://doi.org/10.6084/m9.figshare.26947723]. Due to their large file size, FRAP data will be available without restrictions upon request from the corresponding authors. Source data are provided with this paper.

## Code availability

The computer code used in this work for the model has been deposited in Zenodo [https://doi.org/10.5281/zenodo.13737136] and is also available on Github [https://github.com/davidmrutkowski/Cdc42ParticleFlows].

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

## Acknowledgements

We thank Laetitia Michon for help with molecular biology. Work in SGM's lab is supported by Swiss National Science Foundation grants 310030_191990 and 310030_207909, and European Research Council Advanced grant 101019630 (SexYeast). DV and DMR were supported by NIH grant R35GM136372. Portions of this research were conducted on Lehigh University's Research Computing infrastructure partially supported by NSF Award 2019035 and on Expanse Cluster at SDSC through

allocation TG-MCB180021 and BIO230116 from the Advanced Cyberinfrastructure Coordination Ecosystem: Services & Support (ACCESS) program, which is supported by NSF grants #2138259, #2138286, #2138307, #2137603, and #2138296.

## Author contributions

S.G.M. and D.V. conceived the project and acquired funding. V.V. did all the yeast molecular genetics. S.G.M. did all the microscopy and associated quantifications. D.M.R. and D.V. designed computational models and FRAP analysis methods. D.M.R. performed FRAP analysis, wrote software, and performed simulations. S.G.M., D.M.R., and D.V. wrote the manuscript.

## Competing interests

The authors declare no competing interests.
