## [Peer Review File · Nature Communications]

Cdc42 mobility and membrane flows regulate fission yeast cell shape and survivalREVIEWER COMMENTS

Reviewer #1 (Remarks to the Author):

The authors examine the role of effective membrane flows induced by exo-/ and endocytosis on the stability of the formation of polar Cdc42 domains, vital for cell division in fission yeast. They combine experimental studies of yeast cells modulating membrane binding dynamics of Cdc42 (1ritC, 2ritC, and 3ritC) with a mesoscale particle-based reaction-diffusion model. Within the framework of the proposed model, the authors provide a rationale for the experimentally observed behavior.

The primary mechanism, namely that proteins with strong membrane binding affinity experience an effective outflux from sites with predominant exocytosis, has already been studied previously by the same group in [V. Gerganova, I. Lamas, D. Rutkowski et al., *Sci. Adv.* 7, eabg6718 (2021), manuscript ref. 22]. It is imperative that the authors explain the significant advances in their current work compared to this earlier study. Notably, the previous study already includes all the relevant mechanisms and even incorporates experiments with the Cdc42-xritC constructs like the ones presented in this manuscript.

In terms of modeling, the only novelty seems to be that the authors now included the apparent mechanism of detachment of proteins into the cytoplasm, where they assume infinite diffusivity. This may clarify some details of the initially presented mechanism, but in the absence of qualitative differences, we wonder if this represents a sufficiently significant novelty. Compared to their earlier study (Ref. 22), this manuscript appears to be only an incremental next step, which is noteworthy but not broad enough for the intended readership of *Nature Communications*.

Another serious concern is that there is a major overlap between the present manuscript and Ref. 22: Several figures are reproductions (e.g., 1C, 1D, S1A, and S1B), some without any indication, and entire blocks of text are one-to-one copies. The novel results and differences to Ref. 22, if present, are unclear to us. There is also a significant lack of details in the description of the model and the numerical simulations of the latter. This renders reproducing the study almost impossible.

In addition to the above concerns, we have the following questions and requests:

- The reproduction of figures from other publications should be indicated.
- The authors introduce the concept of 'mobility' as a combination of membrane diffusion and detachment with subsequent fast cytoplasmic diffusion. These two contributing effects should be more clearly disentangled.
- It is unclear whether the proposed shifting of agents after exo-/endocytosis events captures the correct behavior of the system as one neglects volume and surface extension and the resulting dilution. It is necessary to properly include patch generation and dissociation by endo-/exocytosis in the model.

- The model description needs more clarification in the main text and the Methods section. In particular, equations for the reaction kinetics and the functional dependence of the displacements $\Delta r(s)$ should be given.
- We'd like to ask the authors to discuss in more detail the effect that gives rise to the effective membrane flux, namely the overall predominant increase in membrane area due to exocytosis. To some extent, this is obscured by the different values for endo-/exocytosis areas and corresponding rates.
- The authors should describe more precisely how cytoplasmic-membrane particle reactions are implemented.
- The authors should show equations corresponding to Figure 1A.
- We ask the authors to provide more details on the numerical simulations and access to the code to make the study reproducible.
- How is the time step in the Gillespie algorithm related to the dynamics in the Brownian dynamics simulation?
- We ask the authors to comment on the implied dependence of the membrane binding rate of Cdc42-GDP on the dissociation rate constants of Cdc42-xritC-GTP/-GDP stated on page 11.
- The authors mainly describe the observed behavior for the stability of the Cdc42-ATP polar domain and the GAP depletion experiments. Additionally, a mechanistic explanation based on the proposed model would be desirable.
- Throughout the introduction paragraph, various statements lack proper reference. For instance:
 - o 'Polarized secretion, the directional and selective release of molecules from a cell, is critical in many biological processes, including cell communication, tissue development, and immune response. In fungi and other walled cells, it underlies polarized cell growth, as it is necessary for the local delivery of cell wall remodeling and membrane material. Yeast model organisms [...]',
 - o 'The Cdc42 polarity system in yeast is believed to represent one of the best examples of biological pattern formation on cell membranes [...]',
 - o 'Cdc42 accumulation is thought to be driven by the slower mobility of Cdc42-GTP than Cdc42-GDP at the plasma membrane.'
- o The whole paragraph 'Prior modeling work ...' on p. 3

In summary, we have strong reservations regarding the originality and potential broad interest of this work.

Reviewer #2 (Remarks to the Author):

This research article presents a theoretical and experimental study of how cell polarization regulated by Cdc42 influences zones of polarized secretion and associated membrane flows. Spontaneous symmetry breaking during the polarization of yeast (*Saccharomyces cerevisiae* and *Schizosaccharomyces pombe*) has been studied extensively in the past and is a topic of considerable interest and debate. The noteworthy result here is the clear demonstration that changes in *cdc42* mobility directly affect polarized secretion. This is an interesting contribution that is well-written and ought to be published after the authors have considered my suggestions.

I would like the authors to address the following points.

1. The authors have previous work on membrane flows, notably cell patterning by secretion-induced plasma membrane flows. The paper would be strengthened by clearly addressing the differences/additions to their and other authors' previous findings like in Jose M et al. *J Cell Biol*, 2013. Their new finding pertains to how secretion is affected by a modulation of the mobility of Cdc42. A detailed discussion of the model's assumptions and its relationship to prior work is needed to clarify the novelty of the proposed mathematical model with published research. This paper should, at a minimum, cite and discuss the following papers looking at membrane flows during yeast polarization:
2. As the cell tries to establish a single polarization cap from multiple potential polarization sites, which fail to progress, it grows isotropically. How does this isotropic growth (and the presence of multiple polarization sites and potential interaction between them) impact the membrane flows?
3. The geometry for the simulations seems to be spherical. Have the authors tried to change the geometry to understand membrane flows? How do geometry and the cell wall affect the membrane flows? There have been recent theoretical models (Trogon M. et al. *Plos CB* 2018 and Banavar SP. et al. *Plos CB* 2021) looking at how geometry affects polarization during growth. These studies along with possible effects of the geometry and cell wall growth should be discussed in more detail.
4. The authors state: "The spatial probability distributions for exocytosis and endocytosis are convolutions of Cdc42-GTP particle positions with a Gaussian of width w_{exo} and w_{endo} , which we calculated by comparing tip distributions of CRIB-GFP (marking Cdc42-GTP) to profiles of exocytosis and endocytosis in wild type cells." Did the authors map the sites of exocytosis and endocytosis by looking at molecules involved in the exocyst complex such as Exo70 and WASp, implicated in clathrin mediated endocytosis?

Reviewer #3 (Remarks to the Author):

This paper extends the findings from the authors' recent work (Gerganova et al) demonstrating that membrane flow clears slow-mobility membrane proteins from the polarity site in fission yeast. Whereas the previous work developed a modeling strategy for membrane flows but did not model polarity per se, this paper presents a combination of a Cdc42 polarity model (Pablo et al.) with the vesicle traffic/slow GAP model (Gerganova et al). Overall, this model seems well executed, and allows the authors to make some non-obvious predictions that are then confirmed by experiment. The main prediction is that slowing Cdc42 mobility can allow membrane flows to perturb and eventually destroy the polarized Cdc42 distribution. In general, one expects slowing the mobility of a polarity factor to make it easier to retain a polarized distribution of that factor. The point that coupling polarity to membrane flows can reverse this intuition seems important and well validated for the parameter space they explored, and the fission yeast situation.

Overall, I was persuaded by the authors' arguments. But I was confused by some issues listed below. The first in particular seems very important to clarify: I am not persuaded that the FRAP fit approach they use is valid for the tip FRAP, or that a model parameter from that fit makes any sense.

1. I am confused by the reaction modeled with parameter rT . This appears to represent detachment of Cdc42-GTP from the membrane combined with GTP hydrolysis to yield cytoplasmic Cdc42-GDP. No such reaction was present in Pablo et al. While there is prior evidence for membrane detachment of Cdc42-GTP (which should be cited if the reaction is retained), that is thought to be much slower than membrane detachment of Cdc42-GDP (especially because GDI preferentially binds Cdc42-GDP). Yet here it occurs at the same rate. Why is this reaction present and how does its magnitude affect model performance?

The parameter rT is said to be estimated by fitting FRAP recovery of fluorescent Cdc42 at the cell tip, and it was my understanding that the fitted model includes only diffusion and detachment of Cdc42-GTP, with no consideration that Cdc42 detachment may occur through sequential GTP hydrolysis and detachment of Cdc42-GDP. Is that correct? If so, would the FRAP be equally well fitted by adjusting k_{hydro} and rD instead of including rT ?

To me, it would make sense to remove the rT reaction, unless there is independent support for that reaction. If this does not affect the main conclusions, then removing the reaction seems prudent. (Fig. 1F would seem to indicate that a rapid rT is partially responsible for the loss of polarity upon sGAP removal, but the overall conclusion that without sGAP polarity is mostly lost would remain.) If the rT reaction does materially affect the main conclusions, then I would like to see some support for that reaction (and its magnitude) presented and justified.

2. The first paragraph of the Discussion says that "our findings...demonstrate that...differential dynamics of fast Cdc42 and slow inhibitor are critical for pattern

formation". This does not seem quite right to me. They did not examine whether increasing khydro (analogous to Rga3 overexpression, adding fast GAP/inhibitor) would compensate for the loss of slow GAP: why wouldn't this fast-Cdc42/fast-inhibitor scenario work, even with some membrane flow? At minimum, such simulations would be needed before reaching the conclusion that differential dynamics of this specific type are critical.

3. Later in the Discussion they conclude that "establishment of a polarized patch requires its components to turnover faster than the surface on which it is formed". This is an intuitively appealing formulation consistent with their data and model. But it does raise the question of how hyphal fungi like *Neurospora crassa*, which have dramatically faster membrane turnover at the tip than *S. pombe*, can manage to retain polarity sites. Must they have turnover of polarity components at rates that are orders of magnitude faster? Is that realistic? I would be curious to know how the authors think that this might be achieved. (A similar problem was pointed out by Savage et al 2012).

4. In the Methods section on how they did the averaging to get particle concentration profiles, they refer to the "largest cluster" of Cdc42, and to the changing calculation of where the centroid is. It would be good to know how many clusters of Cdc42 are produced, and with what distribution of sizes and lifetimes. Does the model usually have an obvious dominant cluster with rare satellites, or are multi-cluster states common? Also, how stable is the cluster centroid position? If there is usually an obvious dominant cluster, on what timescale does the centroid change position, and by how much?

5. There seemed to be contradictory statements on how Cdc42-GTP diffusion and detachment were modeled for the ritC constructs. In the Results it says "we assumed that the ...Cdc42-GTP...parameters are limited by the ritC anchor and set them the same as for Cdc42-GDP". This would seem to imply that GTP- and GDP-bound forms now share the same mobility. But in the Methods it says they "lowered the corresponding values of Cdc42-GTP ... so that Cdc42-GDP would always have a greater than or equal mobility compared to Cdc42-GTP". I don't understand how both statements can be true. I would assume the Methods version is correct?

6. For the simulations of diploid cells, particle numbers were doubled. But was the sphere volume also doubled? If not, aren't these simulations of overexpressing cells rather than diploids?

7. It was not clear which combinations of GAP deletions (other than the triple) were able to suppress the 3x ritC Cdc42 lethality. Do single GAP deletions suppress? If so, which ones? Do they look any better than the triple? Simulation of GAP deletion cells is described only for the sGAP: what happens if one lowers khydro instead?

8. There is a typo in the Exo/endocytosis Methods where it says "wexo or wexo, respectively". Presumably one of those is intended to be wendo.

We thank the reviewers for their careful reading of our work and their insightful comments. We have addressed them through substantial modelling work and some additional experiments. Detailed responses are provided below between individual reviewers' points.

Reviewer #1 (Remarks to the Author):

The authors examine the role of effective membrane flows induced by exo-/ and endocytosis on the stability of the formation of polar Cdc42 domains, vital for cell division in fission yeast. They combine experimental studies of yeast cells modulating membrane binding dynamics of Cdc42 (1ritC, 2ritC, and 3ritC) with a mesoscale particle-based reaction-diffusion model. Within the framework of the proposed model, the authors provide a rationale for the experimentally observed behavior.

The primary mechanism, namely that proteins with strong membrane binding affinity experience an effective outflux from sites with predominant exocytosis, has already been studied previously by the same group in [V. Gerganova, I. Lamas, D. Rutkowski et al., Sci. Adv. 7, eabg6718 (2021), manuscript ref. 22]. It is imperative that the authors explain the significant advances in their current work compared to this earlier study. Notably, the previous study already includes all the relevant mechanisms and even incorporates experiments with the Cdc42-xritC constructs like the ones presented in this manuscript.

In terms of modeling, the only novelty seems to be that the authors now included the apparent mechanism of detachment of proteins into the cytoplasm, where they assume infinite diffusivity. This may clarify some details of the initially presented mechanism, but in the absence of qualitative differences, we wonder if this represents a sufficiently significant novelty. Compared to their earlier study (Ref. 22), this manuscript appears to be only an incremental next step, which is noteworthy but not broad enough for the intended readership of Nature Communications.

The previous study (Gerganova et al, Sci Adv 2021) established the basis of in-plane membrane flows, showing that polarized secretion coupled to endocytosis over a broader area causes net membrane flow away for the site of secretion and that membrane-associated proteins with long enough residence time at the membrane couple to this flow and are displaced from sites of secretion. We did not address – neither through modelling, nor through experiments – how Cdc42 molecules themselves may be influenced by the flow. On the modelling side, we assumed a stable site of secretion and only looked at downstream consequences. On the experimental side, contrary to what the reviewer states, we did not perform any experiments where we modified Cdc42 membrane binding. Our constructs with 1/2/3-ritC involved either GFP or CIBN and were used as read-outs of membrane flows.

The advance here is to probe how Cdc42 molecules, whose activity establishes the flows, are themselves affected by membrane flows, i.e. we no longer simply consider flows as a downstream consequence of local Cdc42 activity, but as an element that feeds back to Cdc42, by both modifying the distribution of Cdc42 itself and its negative GAP regulators, and can thus modify its own source. In terms of modelling, this is a significant advance as it integrates Cdc42 symmetry-breaking feedback system (modeled as reacting particles) on a flowing biophysical support. Experimentally, we show that the system breaks down when Cdc42 mobility is decreased. This is a clear demonstration that the Cdc42 polarity system can only be maintained through turnover of Cdc42 that is faster than the underlying support.

We have now been more explicit on these advances in the introduction to the manuscript.

Another serious concern is that there is a major overlap between the present manuscript and Ref. 22: Several figures are reproductions (e.g., 1C, 1D, S1A, and S1B), some without any indication, and entire blocks of text are one-to-one copies. The novel results and differences to Ref. 22, if present, are unclear to us. There is also a significant lack of details in the description of the model and the numerical simulations of the latter. This renders reproducing the study almost impossible.

We are sorry for the confusion. This apparent overlap is due to our use of results from our prior work as the basis and motivation for our new work. We re-used a few illustrations from Gerganova et al, with modifications, as a starting point to introduce this new study.

The full model is described by Fig. 1A, 1B, 1C where only part 1C of the model is the same as in Gerganova et al., hence the use of this panel. In our technical description of the part of the model that corresponds to Fig. 1C, we had tried to condense a more detailed description in Gerganova et al. but we understand this came out as both cut-and-paste and lack of detail.

Fig. 1D shows our new measurements of Cdc42 mobilities in relationship to the depletion regions discovered in Gerganova et al.

We did not intend to present S1A and S1B as new results but rather added them to help the readers with some key background observations so that they don't have to extract them from Gerganova et al.

The above are now stated much more explicitly in the legends. We also contacted Science Advances to confirm we have permissions for re-use. We paste here the reply from the journal: "Per the Science Advances Author License to Publish, after publication of your Science Advances paper, no further permission is needed from AAAS in order for you to use/modify those figures & tables in a new work you write (eg: book chapter, journal article, dissertation, blog post etc.). This policy applies to your use in both non-commercial and commercial works, so long as you credit the original Science Advances paper and indicate the type of CC license it was published with."

We have added additional details in the description of the model and numerical simulations in response to specific reviewer comments (see below). As originally intended, we have added a Code availability section to the manuscript and the code is now available at <https://github.com/davidmrutkowski/Cdc42ParticleFlows>, allowing full reproducibility.

In addition to the above concerns, we have the following questions and requests:

- The reproduction of figures from other publications should be indicated.

This is now done in the legends.

- The authors introduce the concept of 'mobility' as a combination of membrane diffusion and detachment with subsequent fast cytoplasmic diffusion. These two contributing effects should be more clearly disentangled.

We understand the reviewer's concern and we have clarified in the text where there may be confusion. We still find it convenient to use a single term for brevity purposes. However, we hope it is clear that we did try to characterize and resolve the separate contributions of membrane diffusion and membrane dissociation/association. There is a difference between these two mobility modes as diffusion tends to broaden concentration peaks on the membrane over time while dissociation

followed by fast cytoplasmic diffusion and re-association leads to a gradual reduction of a membrane concentration peak height and increase in the surrounding values outside the peak, without broadening of the peak. Simulations also show that these two mobility mechanisms are not entirely equivalent (e.g. Figure 3C where the system behavior is generally different along the two bands). However, the difficulty in resolving these two modes of protein mobility is suggested by our fits to FRAP results, which generally have two best-fit bands at a finite koff / zero diffusion coefficient and at a finite diffusion coefficient / zero koff (e.g. Gerganova et al, Sci Adv 2021, Fig S4). To address this issue and trim down these bands as much as possible, we averaged fits to FRAP for bleaches of different widths, which have different sensitivity to detachment and diffusion.

- It is unclear whether the proposed shifting of agents after exo-/endocytosis events captures the correct behavior of the system as one neglects volume and surface extension and the resulting dilution. It is necessary to properly include patch generation and dissociation by endo-/exocytosis in the model.

We acknowledge that our simulations capture only a steady-state situation and not the full events of growth. We note that the timescales of surface and volume growth, which lead to cell shape changes, are much longer than the timescales of the molecular reactions we simulate, which provides support for our approach. We also note that membrane remodelling (i.e. local surface extension) is not neglected, as molecules are displaced as consequence of individual exo-/endocytic events.

However, to address the reviewer's concern, and to further explore the assumptions of our model, we created new simulations that include volume and surface expansion. In fission yeast, the spherocylindrical geometry is critical in accounting for volume expansion through cell elongation, therefore we implemented particle-based reactions along a growing cell tip. One major difficulty in producing simulations that reflect realistic growth in volume and surface is that growth is a consequence of Cdc42 local activity. Thus, the shape of the simulated surface/volume should in principle evolve as a consequence of the formation of the Cdc42-GTP patch. This poses substantial difficulties, as no element of our simulations ensures that zones of Cdc42 activity should coincide with the tip of a simulated spherocylindrical shape. Indeed, previous work has shown that the shape of the cell may by itself cause instability in the position of the Cdc42 patch (Trogon et al, PLoS Comp Biol 2018), which can be stabilized through mechanical feedback (Banavar et al, PLoS Comp Biol 2021). Thus, implementing growth on a self-polarizing system is not trivial.

To simplify the problem, rather than implementing membrane flows based on discrete events for endocytosis and exocytosis defined by the position of Cdc42-GTP, we used an average, azimuthally-symmetric profile of membrane flow centered at the cell tip. Working in the reference frame where the cell tip is at rest, in these simulations each particle is displaced radially along the spherical cap or cylindrical segment according to an average velocity that accounts for cell growth and net membrane area added or removed. This velocity flow profile, shown in Figure R1, was determined by: (1) Freezing Cdc42-GTP particles in a patch with width equal to the average patch from steady state WT/1ritC simulation on the sphere surface as in Figure 1E and 3A. (2) Running the simulation with endocytosis and exocytosis events generated by the stationary Cdc42-GTP and calculating the induced flow on probe ghost particles placed uniformly on the sphere. The ghost particles were used to accumulate the average flow and were not displaced during these simulations.

Figure R1. Average flow velocity away from the north pole due to random exo/endocytosis events under growth conditions occurring for the WT (red) and 1ritC (blue) distribution of Cdc42-GTP

The spherocylinder simulations begin with the same surface density and number of particles as our sphere simulations. We initialize the cylinder length to be $3.0 \mu\text{m}$ with the spherical radius remaining at $1.8 \mu\text{m}$ as in all other simulations. For each timestep we determine the displacement on the spherocylinder surface due to the flow velocity in Figure R1 and diffusion. We first displace the particle on the surface it begins on (e.g. hemisphere or cylinder) and then if the particle would cross the boundary between two surfaces, we advance the particle up to this boundary and continue the displacement on the other surface. Reflecting boundary conditions were applied at the free edge of the cylindrical segment. We performed simulations where the reflecting boundary was moved at a rate equal to the rate of tip growth, or else kept stationary. Since cell growth is relatively slow compared to the time of steady patch formation, we kept the number of particles fixed throughout the simulation (Figure R2). To compare the spherocylinder geometry results, both with or without explicit growth, to results using spherical geometry, we implemented the same algorithm for a sphere, assuming the flow originates at a fixed location on the sphere.

Figure R2. Spherocylinder growth simulations where cylinder surface ends in a reflecting boundary (WT case). The shifting downwards of this boundary is associated with net cell growth (net elongation rate of $0.031 \mu\text{m}/\text{min}$). All sGAP particles are initiated in the cylinder portion of the surface to avoid transient patch formation outside the tip region.

We find that all three cases (growing spherocylinder, static spherocylinder, and sphere) lead to a steady state patch of similar width and concentration amplitude to each other (Figure R3A), as well as to the simulations of Figure 1E and 3A. Because we initialized the sGAP outside the top hemisphere, we observe the same lateral accumulation in sGAP concentration at $\sim 3.5 \mu\text{m}$, as seen in Fig. 1E, at much later times (Figure R3B). For these later times, the Cdc42 patch weakens in the case with spherocylinder growth due to growth-induced particle dilution (Figure R3B).

The observation that different geometries do not influence the size of the patch in these simulations demonstrates that our sphere simulations can be used as a good approximation for polarization of a growing spherocylinder. We added these three figures as new Supplemental Figure 2 and added a few sentences in the main text and a subsection in the Modeling methods section describing these spherocylinder simulations.

Figure R3. Concentration profiles comparing polarization under WT and 1ritC conditions for growing spherocylinder, static spherocylinder, and sphere, using particle displacements due to endocytosis and exocytosis as in Fig. R1. Simulations were run for 5000 s. (A) Concentration profile averages at intermediate times after patch formation (400 to 1000 s), as indicated in the panels. (B) Concentration profile averages at longer times (2500 to 5000 s), as indicated in the panels. A concentration peak of sGAP at $\sim 3-3.5 \mu\text{m}$ is observed at long times due to flow displacement. The Cdc42 patch weakens in the case with spherocylinder growth due to growth-induced particle dilution.

- The model description needs more clarification in the main text and the Methods section. In particular, equations for the reaction kinetics and the functional dependence of the displacements $\Delta r(s)$ should be given.

We have added the model reactions in a table (new Table S1) and the functional dependence of the displacement $\Delta r_{exo/endo}(s)$ to the Mathematical Modeling Methods section. We list the model reactions in a table, following earlier work using particle simulations [Pablo, Ramirez, Elston *PLOS Comp Biol* 2018]. Mathematically representing our discrete particle reactions in the form of equations would involve writing equations in a field operator representation [Doi *J Phys A* 1976 9:1465, 9:1479], which we find to be unnecessarily complex for the purposes of this work. Since our simulations are not performed at the continuum level, our model does not reduce to partial differential equations such as

$$\frac{\partial Cdc42T}{\partial t} = (\lambda_{SD}Scd + \lambda_{ST}ScdCdc42T) \cdot Cdc42D - k_{hydro}Cdc42T - r_T Cdc42T - \lambda_{GAP}Cdc42T \cdot sGAP - \lambda_{ST}Scd \cdot Cdc42T + r_{ST}ScdCdc42T + \nabla(\mathbf{j}_{Cdc42T}),$$

where the last term represents the flux of the species on the membrane due to the diffusion and flow due to exocytosis / endocytosis events.

- We'd like to ask the authors to discuss in more detail the effect that gives rise to the effective membrane flux, namely the overall predominant increase in membrane area due to exocytosis. To some extent, this is obscured by the different values for endo-/exocytosis areas and corresponding rates.

The effective membrane flux away from the tip is driven in part by the difference in net area addition/removal and in part by the wider profile of endocytosis compared to exocytosis. Indeed, in Gerganova et al. we showed that having neither of these aspects ($\sigma_{exo} = \sigma_{endo}$ and no net growth) leads to no strong depletion of inert particles from the tip region (Fig. S2C in that work). Both aspects contribute additively to the overall displacement. In the model presented in this work, there is an overall net addition of membrane since exocytosis delivers $0.0214 \mu\text{m}^2/\text{s}$ while endocytosis removes $0.0157 \mu\text{m}^2/\text{s}$. This difference derives from the fact that we are modeling a growing tip of *S. pombe* which grows at a rate of $20.7 \mu\text{m}^2/\text{h}$ as outlined in Gerganova et al. We modified parts of the first section of the Results section to make these points clearer as follows: "Membrane flux is due in part to the width of the exocytosis profile being narrower than that of endocytosis (Gerganova et al. Sci Adv 2021), and in part to the higher rate of exocytosis than endocytosis ($r_{exo}A_{exo} = 0.0214 \mu\text{m}^2/\text{s}$ while $r_{endo}A_{endo} = 0.0157 \mu\text{m}^2/\text{s}$) leading to net cell growth. For simplicity we report results for flows on the surface of a sphere without explicitly simulating growth, assuming the center of a Cdc42 patch corresponds to the frame of reference of a growing tip".

- The authors should describe more precisely how cytoplasmic-membrane particle reactions are implemented.

Unimolecular cytoplasmic-membrane particle reactions are implemented as described in the *Particle reactions* sub-section of the Methods section where a particle in the cytoplasm is placed anywhere on the sphere surface (fast cytoplasmic diffusion assumption) with equal probability with a rate according to its binding rate as given in Table 1. Bimolecular cytoplasmic-membrane particle reaction only involves cytoplasmic Scd1/2 reacting with membrane bound Cdc42-GTP. We have added the following sentence to the *Mathematical Modeling Methods* section to describe how the final position of this resulting complex is determined: "For the reaction between cytoplasmic Scd1/2 and membrane-bound Cdc42-GTP we place the resulting complex at the position the Cdc42-GTP had since it is the only particle on the membrane for this reaction. Dissociation from the membrane places the particles in the cytoplasmic pool."

- The authors should show equations corresponding to Figure 1A.

As mentioned above, we represented the reactions as a list in new Table S1.

- We ask the authors to provide more details on the numerical simulations and access to the code to make the study reproducible.

We have added additional details in the description of the model and numerical simulations. The code is now available at <https://github.com/davidmrukowski/Cdc42ParticleFlows>.

- How is the time step in the Gillespie algorithm related to the dynamics in the Brownian dynamics simulation?

All reactions are performed with the same discrete timestep as for the Brownian diffusion dynamics ($\Delta t = 0.001$ s), except for exocytosis/endocytosis and unimolecular membrane binding reactions that were based on a varying timestep determined by the Gillespie algorithm, independent of this discrete timestep. The timesteps determined from the Gillespie algorithm occur in between discrete timesteps as shown in the timeline in Figure R4. We use the Gillespie algorithm for exo/endocytosis events that occur less frequently. Since membrane binding reaction rates do not depend on spatial position, and could also occur rarely (such as for Cdc42-xritC3), we also used a Gillespie algorithm for these reactions. As a result, for WT cells, both the Gillespie and discrete timesteps are typically around the same value. Diffusion for every particle in the system occurs to advance the system up to discrete timesteps and to times associated with exo/endocytosis. To handle the split between Gillespie and discrete timestep, each particle has two times associated with it; t_{react} , which is the most recent time the particle was allowed to react up to, and t_{pos} which gives the most recent time the particle was diffused up to. Specifically, the probability to react is determined by $P_{react} = 1 - \exp[-r(n\Delta t - t_{react})]$, where n is the number of discrete time steps. Particles are diffused over the time period of $t - t_{pos}$, where t is the current simulation time. Upon membrane binding of a specific particle, both t_{react} and t_{pos} are set equal to the current simulation time. We have added these clarifications in the *Time Step* section of the *Mathematical Modeling Methods* section.

Figure R4. Timeline showing placement of discrete timestep events ($n\Delta t$) and events that occur at times determined using the Gillespie algorithm (dashed lines).

- We ask the authors to comment on the implied dependence of the membrane binding rate of Cdc42-GDP on the dissociation rate constants of Cdc42-xritC-GTP/-GDP stated on page 11.

We have clarified this equation in the main text in the *Mathematical Modeling Methods* section (*Assumptions of particle reaction scheme* section):

“The membrane binding rate of Cdc42-GDP, k_D , was set dependent on the dissociation rates r_T , r_D of Cdc42 GTP/GDP so that approximately 3/5 of Cdc42 is on the membrane. We determined the functional form of this dependence from the steady state balance equation $k_D N_{cyto} = N_T r_T + N_D r_D$, where N_{cyto} , N_T , N_D are the number of Cdc42 in the cytoplasm, and bound to the membrane with GTP or GDP, respectively. Additionally, we fix $N_D = 2\beta N_{cyto}$ and $N_{cyto}/(N_{cyto} + N_D + N_T) = 1/(1 + 3\beta)$, where for our simulations $\beta = 0.513$ in order to have a ratio of 3/5 of the total Cdc42 on the membrane. Solving these three equations leads to the form shown in Table S2: $k_D = \beta(r_T + 2r_D)$. Indeed, for the reference WT-Cdc42 simulations $N_D \approx 2200$, $N_{cyto} \approx 2220$, $N_T \approx 380$, and $N_{scd12T}^{WT} \approx 200$ ”.

We note that this calculation of k_D assumes that the binding affinity of Cdc42-xritC constructs for the plasma membrane remains approximately the same as for WT cells. We checked that our results do not depend on this assumption as shown below in Figure R5 (also provided as Figure S10 in the

manuscript). The main difference is an overall higher level of Cdc42 on the membrane when using a constant value of $k_D = 0.1 \text{ s}^{-1}$ for all three Cdc42-xritC constructs (compare Figure R5A to R5B).

Figure R5 (A) Profiles for ritC conditions at steady state taken from Figure 3A (B) Profiles from simulations where the membrane binding rate of Cdc42-GDP, k_D , was fixed at a value of 0.1 s^{-1} .

- The authors mainly describe the observed behavior for the stability of the Cdc42-ATP polar domain and the GAP depletion experiments. Additionally, a mechanistic explanation based on the proposed model would be desirable.

We have modified our Discussion section slightly to focus more on how the model generates patches both with the discrete GAP molecule and without it as shown below:

“Our findings, supported by experimental validation, demonstrate that differential dynamics of fast Cdc42 and slow inhibitor provide a mechanism for pattern formation in the presence of membrane flow. In this mechanism, membrane flows enable protein dispersion that both constrains the membrane affinity of polarity factors and allows long-range displacement of inhibitors, implementing a transport-driven lateral inhibition mechanism as in classic activator-inhibitor pattern formation. While prior studies showed that Cdc42-GTP relative low mobility traps it within its activation region, our study, as discussed below, reveals that Cdc42 diffusion coefficient on the membrane, and membrane dissociation rate, should nevertheless be fast enough for the average duration of Cdc42 within the active zone to be shorter than the characteristic time for membrane flow to displace it, in order to generate a zone with clearly defined boundaries.

[...] While membrane flows promote polarization through removal of negative factors, they also destabilize the polarity patch, which replenishes through rapid Cdc42 flux, in agreement with previous modelling effort. We now show, both in silico and experimentally, that sufficiently fast dynamics (fast membrane unbinding rates and/or fast lateral diffusion) of Cdc42 is required for efficient polarization. The loss of polarization upon decreased Cdc42 mobility can be understood in the following way: Arrival of secretory vesicles at the patch causes local displacement and dilution of Cdc42, leading to widening and weakening of the patch. Consequently, secretory vesicle delivery is less polarized, reducing the

net membrane flow. sGAP is thus less efficiently displaced, causing stronger overlap between sGAP and Cdc42-GTP and consequent Cdc42 inactivation. By contrast, fast dynamics allow Cdc42 to rapidly repopulate the zones of secretion, alleviating the effects of the membrane turnover.[...] We conclude that establishment and maintenance of a polarized patch requires its components to turn over faster than the surface on which it is formed.

In principle, a high Cdc42-GTP turnover at the membrane can result not only from fast Cdc42 protein mobility, but also from a high rate of GTP hydrolysis. Thus, an alternate mechanism for patch formation in the presence of flows that displace Cdc42, but in absence of sGAPs, may involve a higher level of k_{hydro} (equivalent to Rga3 overexpression compared to our reference case, and provided that Cdc42-GDP mobility remains sufficiently high)."

- Throughout the introduction paragraph, various statements lack proper reference. For instance:
 - o 'Polarized secretion, the directional and selective release of molecules from a cell, is critical in many biological processes, including cell communication, tissue development, and immune response. In fungi and other walled cells, it underlies polarized cell growth, as it is necessary for the local delivery of cell wall remodeling and membrane material. Yeast model organisms [...]',
 - o 'The Cdc42 polarity system in yeast is believed to represent one of the best examples of biological pattern formation on cell membranes [...]',
 - o 'Cdc42 accumulation is thought to be driven by the slower mobility of Cdc42-GTP than Cdc42-GDP at the plasma membrane. '
 - o The whole paragraph 'Prior modeling work ...' on p. 3

We have added references where they were missing in these sections.

In summary, we have strong reservations regarding the originality and potential broad interest of this work.

We hope we have satisfactorily explained the advance of this study. In terms of originality and broad interest, this is the first study that examines in detail how membrane flows induced by polarized secretion modulate the establishment of the polarity domain. The concept that the proteins that define the polarity patch need to exhibit faster mobility than the membrane support on which they assemble is a broadly valid one, with direct relevance to understand polarized cell growth in any organism.

Reviewer #2 (Remarks to the Author):

*This research article presents a theoretical and experimental study of how cell polarization regulated by Cdc42 influences zones of polarized secretion and associated membrane flows. Spontaneous symmetry breaking during the polarization of yeast (*Saccharomyces cerevisiae* and *Schizosaccharomyces pombe*) has been studied extensively in the past and is a topic of considerable interest and debate. The noteworthy result here is the clear demonstration that changes in *cdc42* mobility directly affect polarized secretion. This is an interesting contribution that is well-written and ought to be published after the authors have considered my suggestions.*

I would like the authors to address the following points.

1. The authors have previous work on membrane flows, notably cell patterning by secretion-induced plasma membrane flows. The paper would be strengthened by clearly addressing the differences/additions to their and other authors' previous findings like in Jose M et al. J Cell Biol, 2013. Their new finding pertains to how secretion is affected by a modulation of the mobility of Cdc42. A detailed discussion of the model's assumptions and its relationship to prior work is needed to clarify the novelty of the proposed mathematical model with published research. This paper should, at a minimum, cite and discuss the following papers looking at membrane flows during yeast polarization:

There are two major differences between our current work and earlier work, including our previous Gerganova et al, Sci Adv 2021 and the Jose et al, JCB 2013 paper cited above. First, we explicitly model the effect of membrane flows – i.e the remodeling of the plasma membrane by exocytic and endocytic events – on the displacement of Cdc42 proteins, and probe the importance of Cdc42 mobility at the membrane. Second, we explicitly model the effect of Cdc42 GAP proteins, which are themselves displaced from the polarity site by the flow. Our work comes to the conclusions that Cdc42 needs to turn over faster than the rate of membrane flow and that the flow-mediated GAP displacement promotes the formation of a polarized patch.

Most prior modelling efforts did not take membrane flows into account: Jose et al, JCB 2013 models how endocytosis can occur around zones of exocytosis, where polarity factors are simply dispersed from sites of delivery at secretion sites through diffusive motion; Gerganova et al, Sci Adv 2021, modelled the effect of membrane flows on the distribution of membrane-associated proteins, but assumed a static site of secretion itself unaffected by flows. To our knowledge, only two prior models explored the effect of polarized secretion on cell polarity establishment by simulating new membrane insertion at the plasma membrane: 1) Layton et al, Curr Biol 2011, concluded that polarized secretion perturbs cell polarity by diluting the Cdc42 patch even upon delivery of Cdc42 by secretory vesicles; 2) Okada et al, Dev Cell 2013 modelled a growing surface displacing septins from the site of secretion defined by Cdc42-GTP, leading to the formation of a ring of septins, which were proposed to recruit Cdc42 GAPs to feedback and restrict Cdc42 activity and growth of the bud in its center. This second study used a continuum concentration approximation rather than a particle-based method, and did not introduce endocytosis nor the concept of lateral membrane flow. Our results agree with these previous conclusions (weakening of the polarity patch by secretion, and exclusion of slow-mobile proteins such as septin filaments), which we also extend by showing a direct regulation through exclusion of GAP proteins and probing the importance of Cdc42 fast mobility. All these papers are cited in our manuscript.

2. As the cell tries to establish a single polarization cap from multiple potential polarization sites, which fail to progress, it grows isotropically. How does this isotropic growth (and the presence of multiple polarization sites and potential interaction between them) impact the membrane flows?

The reviewer asks an interesting question regarding the kinetics of approach to steady state in our model, and the impact of membrane flow. Quantifying these kinetics is somewhat complex and so we feel it goes beyond the scope of this work. We can however highlight some key aspects of this behavior. Indeed, during symmetry breaking in the model, multiple polarization sites can initially form, depending on Cdc42 mobility parameters (please also see Figure R10 in response to reviewer 3).

The evolution of these patches depends on the following dynamical mechanisms:

(1) There is competition between patches due to both protein mobility in/out of patches and the Cdc42 scaffold mediated positive feedback, and smaller patches are expected to weaken against larger ones, similar to reaction-diffusion models without membrane flows.

(2) Patches compete for membrane delivery that displaces sGAP. Since sGAP will infiltrate small patches if there is not enough membrane delivery, this mechanism leads to an enhanced rate at which small patches disappear, leading to the growth of larger patches.

(3) Membrane flows could allow two nearby patches to merge due to their combined membrane flow. The net membrane flow could displace the sGAP from the region between the two patches and generate a region that is devoid of sGAP, allowing the two patches to merge. To investigate this mechanism, we calculated the average membrane flows due to two patches of Cdc42-GTP, as shown below in Fig. R6. The membrane flow between the two patches can be seen to drive particles (that are not exactly along the centerline of the two patches) out of the region between the two patches.

Figure R6. Net membrane flow vectors on sphere surface for two patches of Cdc42-GTP (orange). The flows were calculated by placing static Cdc42-GTP particles (orange) and calculating the induced flow on probe ghost particles on the sphere (green arrows).

For WT simulations we did not observe the appearance of two competing patches, so positive feedback and membrane-flow displacement work simultaneously to establish a single patch. For conditions corresponding to Cdc42-1ritC cells, two clearly distinct patches do form (Figure R10 in response to reviewer 3 below). For this latter case we observe one patch growing at the expense of a shrinking one, without the two moving closer together, indicating that patch competition for Scd1/2 or membrane delivery (mechanisms 1 and 2 above) dominate over membrane-flow-induced lateral merging (mechanism 3) in the process of single patch formation.

We note that the process of patch emergence in the presence of flows cannot in principle be reduced to results of prior works that rely on non-linear positive feedback. In fact, reducing the strength of positive feedback by reducing λ_{STT} in the model leads to a reduction in the overall patch size and strength, but it does not abolish polarization into a single patch (Figure R7 and new supplemental Figure S8). This occurs for both WT and Cdc42-1ritC parameters even for $\lambda_{STT}=0$. For the Cdc42-1ritC case, a single patch emerges directly without an intermediate two-patch competing regime (not shown). We have added a paragraph in the Discussion section that describes this result.

Figure R7. Concentration profiles when reducing non-linear positive term, λ_{STT} , from reference value of 5 s^{-1} , keeping all other parameters same as WT cells.

3. The geometry for the simulations seems to be spherical. Have the authors tried to change the geometry to understand membrane flows? How do geometry and the cell wall affect the membrane flows? There have been recent theoretical models (Trogon M. et al. Plos CB 2018 and Banavar SP. et al. Plos CB 2021) looking at how geometry affects polarization during growth. These studies along with possible effects of the geometry and cell wall growth should be discussed in more detail.

These are interesting questions. We chose the sphere as this represents the shape of the cell pole of the *S. pombe* cells quite faithfully, where, to the best of our knowledge, all exocytic activity occurs. In response to this comment and also to Reviewer 1, we have simulated an azimuthally averaged exo/endocytosis version of our model on the surface of a growing spherocylinder. We kindly refer the Reviewer to our response above. Briefly, we found that essentially identical profiles develop on the surface of the sphere or the spherocylinder (Fig. R3 and new Supplementary Figure S2), which validates our use of the sphere.

The cited studies showed that some models of Cdc42 symmetry breaking were unstable in non-spherical shapes, notably in shapes resembling that of mating cells, i.e. cells with short growth projections. The second of these papers showed that stability could be obtained in these shapes through mechanical feedback between cell wall strain (signaling through Rho GTPase) and the Cdc42 polarization model. Investigating our full model integrating membrane flows in different morphologies is beyond the current study, but it would be interesting to test if the stabilization of the polarity patch by the local depletion of GAPs may also suffice (or contribute) to stabilizing the patch at the tip of cells with short projections. For instance, it is interesting that *S. cerevisiae* mating projection localize septins around their base, which (as proposed in Okada et al, Dev Cell 2013) may be displaced by flows and recruit Cdc42 GAPs. We have added the perspective to probe the role of membrane flows in the discussion.

4. The authors state: “The spatial probability distributions for exocytosis and endocytosis are convolutions of Cdc42-GTP particle positions with a Gaussian of width w_{exo} and w_{endo} , which we

calculated by comparing tip distributions of CRIB-GFP (marking Cdc42-GTP) to profiles of exocytosis and endocytosis in wild type cells.” Did the authors map the sites of exocytosis and endocytosis by looking at molecules involved in the exocyst complex such as Exo70 and WASp, implicated in clathrin mediated endocytosis?

We did that in Gerganova et al, Sci Adv 2021 (in Figure S3). The convolutions used here result in distributions that match the experimental profiles (Figure S4 of the revised manuscript).

Reviewer #3 (Remarks to the Author):

This paper extends the findings from the authors’ recent work (Gerganova et al) demonstrating that membrane flow clears slow-mobility membrane proteins from the polarity site in fission yeast. Whereas the previous work developed a modeling strategy for membrane flows but did not model polarity per se, this paper presents a combination of a Cdc42 polarity model (Pablo et al.) with the vesicle traffic/slow GAP model (Gerganova et al). Overall, this model seems well executed, and allows the authors to make some non-obvious predictions that are then confirmed by experiment. The main prediction is that slowing Cdc42 mobility can allow membrane flows to perturb and eventually destroy the polarized Cdc42 distribution. In general, one expects slowing the mobility of a polarity factor to make it easier to retain a polarized distribution of that factor. The point that coupling polarity to membrane flows can reverse this intuition seems important and well validated for the parameter space they explored, and the fission yeast situation.

Overall, I was persuaded by the authors’ arguments. But I was confused by some issues listed below. The first in particular seems very important to clarify: I am not persuaded that the FRAP fit approach they use is valid for the tip FRAP, or that a model parameter from that fit makes any sense.

1. I am confused by the reaction modeled with parameter rT . This appears to represent detachment of Cdc42-GTP from the membrane combined with GTP hydrolysis to yield cytoplasmic Cdc42-GDP. No such reaction was present in Pablo et al. While there is prior evidence for membrane detachment of Cdc42-GTP (which should be cited if the reaction is retained), that is thought to be much slower than membrane detachment of Cdc42-GDP (especially because GDI preferentially binds Cdc42-GDP). Yet here it occurs at the same rate. Why is this reaction present and how does its magnitude affect model performance?

The parameter rT is said to be estimated by fitting FRAP recovery of fluorescent Cdc42 at the cell tip, and it was my understanding that the fitted model includes only diffusion and detachment of Cdc42-GTP, with no consideration that Cdc42 detachment may occur through sequential GTP hydrolysis and detachment of Cdc42-GDP. Is that correct? If so, would the FRAP be equally well fitted by adjusting r_{hydro} and rD instead of including rT ?

To me, it would make sense to remove the rT reaction, unless there is independent support for that reaction. If this does not affect the main conclusions, then removing the reaction seems prudent. (Fig. 1F would seem to indicate that a rapid rT is partially responsible for the loss of polarity upon sGAP removal, but the overall conclusion that without sGAP polarity is mostly lost would remain.) If the rT reaction does materially affect the main conclusions, then I would like to see some support for that reaction (and its magnitude) presented and justified.

Thank you for this important point. The membrane detachment rate for Cdc42-GTP (rT) we considered was indeed an effective rT , estimated from the FRAP fits, but representing an aggregate between rates

of Cdc42-GTP hydrolysis and detachment of Cdc42-GDP (and perhaps Cdc42-GTP). The reviewer is also correct in that the main conclusions of the model generally remain valid without rT (rT = 0).

Nevertheless, Cdc42-GTP is not irreversibly stuck at the plasma membrane and previous work has shown PM-cytosol exchange of Cdc42-GTP, as also pointed out by the reviewer (notably Woods et al, Curr Biol 2016), which argues against complete removal of the rT reaction. We agree that FRAP fits at cell tips are more difficult than fits from FRAP at cell sides. To estimate a rate for Cdc42-GTP detachment from the membrane, independently of GTP hydrolysis, we have conducted additional FRAP experiments with a GTP-locked allele of Cdc42 (Cdc42-Q61L-mCherry^{SW}) expressed in addition to endogenous Cdc42. As Cdc42-Q61L-mCherry^{SW} localizes ubiquitously at the plasma membrane, we could perform FRAP at the cell side in regions of various sizes and use these to distinguish lateral diffusion from membrane detachment. For these experiments, we kept Cdc42-Q61L expression moderate and only analyzed cells that retained the normal rod morphology. The best fit to this FRAP recovery indicates a much lower, but non-null, value of the dissociation rate for Cdc42-GTP to that of Cdc42-GDP (see updated Figure 1D and S3C). The diffusion coefficient of Cdc42-GTP is not distinguishable from that of Cdc42-GDP within our resolution. This came as a surprise to us, as analysis of half-tip FRAP in our previous work (Bendezu et al, PLoS Biol 2015) had experimentally suggested that the diffusion coefficient of Cdc42 at cell tips, which should be dominated by Cdc42-GTP, is slower by roughly an order of magnitude. In that earlier work we did not have an estimate of the Cdc42-GTP dissociation rate, but we had nevertheless interpreted the fact that Cdc42-GTP is less mobile than Cdc42-GDP by roughly one order of magnitude to be due to slower lateral diffusion of Cdc42-GTP. While the scatter in D and k_{off} fit values in our current fits is significant, overall, our fits on Cdc42-Q61L-mCherry^{SW} FRAPs now indicate that the main difference between Cdc42-GDP and Cdc42-GTP mobility resides in slower detachment rates of Cdc42-GTP. Our former interpretation was partly based on the observation that deletion of the GDI *rdi1Δ*, which is the only characterized mechanism of Cdc42-GDP membrane detachment, causes only modest changes in Cdc42 mobility. The data we now show suggest that there are other modes of Cdc42 membrane detachment, which act preferentially on Cdc42-GDP (in agreement with observations in Woods et al, Curr Biol 2016). We re-ran all simulations with this new reference value of D and k_{off} for Cdc42-GTP where required and have updated Figures 1, 3, 4A and Supplemental Figures 4 and 5 with these new simulations. All simulation results were robust to these changes.

*2. The first paragraph of the Discussion says that “our findings...demonstrate that...differential dynamics of fast Cdc42 and slow inhibitor are critical for pattern formation”. This does not seem quite right to me. They did not examine whether increasing *k_{hydro}* (analogous to *Rga3* overexpression, adding fast GAP/inhibitor) would compensate for the loss of slow GAP: why wouldn't this fast-Cdc42/fast-inhibitor scenario work, even with some membrane flow? At minimum, such simulations would be needed before reaching the conclusion that differential dynamics of this specific type are critical.*

We thank the reviewer for pointing out this issue and agree that our formulation was not optimal. Our data show that the timescale of Cdc42-GTP at the membrane has to be shorter than the characteristic membrane flow to sustain a stable polarity patch. Our data further show that cells can tolerate a substantial decrease in Cdc42 mobility (a substantial increase in Cdc42 membrane residence time) because of the action of slow GAP, which counteracts the reach of its active form, together with positive feedback reactions.

To address the question whether an increase in Cdc42 hydrolysis rate would similarly extend the cell's tolerance to reducing Cdc42 mobility, we varied the value of k_{hydro} in simulations without sGAP (updated Figure 1F and additional cases of Figure R8, also shown in a new supplemental Figure S7

discussed in the Discussion section). There is a range of k_{hydro} values where a distinct Cdc42-GTP patch forms in simulations without sGAP but with a higher value of k_{hydro} than in our reference WT case ($= 0.175 s^{-1}$). At high values of k_{hydro} ($\geq 35 s^{-1}$), the patch becomes unstable because deactivation of Cdc42-GTP occurs too quickly. At low values of k_{hydro} ($\leq 0.0175 s^{-1}$) Cdc42 remains active too long and spreads nearly over the entire sphere surface. This result further supports the finding that the lifetime of a Cdc42-GTP molecule at the membrane needs to be short enough relative to membrane flow for a patch to form. We have changed the quoted statement to the following:

“Our findings, supported by experimental validation, demonstrate that differential dynamics of fast Cdc42 and slow inhibitor provide a mechanism for pattern formation in the presence of membrane flow.”

Figure R8. Simulations at WT parameters without sGAP, as a function of k_{hydro} . The reference value of $0.175 s^{-1}$ and a case with $0.875 s^{-1}$ are shown in Figure 1F. Increase of k_{hydro} by a few times compared to the reference value recovers a more focused patch. Increasing or decreasing k_{hydro} too much results in loss of Cdc42-GTP polarization.

3. Later in the Discussion they conclude that “establishment of a polarized patch requires its components to turnover faster than the surface on which it is formed”. This is an intuitively appealing formulation consistent with their data and model. But it does raise the question of how hyphal fungi like *Neurospora crassa*, which have dramatically faster membrane turnover at the tip than *S. pombe*, can manage to retain polarity sites. Must they have turnover of polarity components at rates that are orders of magnitude faster? Is that realistic? I would be curious to know how the authors think that this might be achieved. (A similar problem was pointed out by Savage et al 2012).

This is an interesting question. To address it, we used estimates of membrane turnover measured for *N. crassa* (Bartnicki-Garcia et al, Fungal Genetics and Biology 2018), which estimate new PM added by exocytosis about 150-fold higher and PM removed by endocytosis about 27-fold higher than in *S. pombe*.

We have calculated the corresponding 50% tip depletion boundaries for the case of 150-fold faster exocytosis and 27-fold higher endocytosis using both our analytical expression $f_1 D/v < L < f_2 v/k_{off}$ (light red) and simulations of passive particles (red line) as in Gerganova et al. (Sci Adv 2021) in Fig R9 below (and new supplemental Figure S9). Here f_1 and f_2 are numerical prefactors of order unity. The corresponding boundaries for *S. pombe* are in blue. For these passive particle simulations, the values of α and γ were both set to 1.0. For the *N. crassa* particle simulations we used a sphere with radius 5 μm , where exocytosis and endocytosis profile standard deviations were multiplied by 2.78 that for *S. pombe*. Additionally, exocytosis and endocytosis rates were multiplied by 150 and 27 that for *S. pombe*, respectively, in order to match to the total rate of membrane addition/removal as given in [Bartnicki-Garcia et al, 2018]. Vesicle size was assumed to be the same as in Table S3. The exo/endocytosis cutoff was also increased to 5 μm for the *N. crassa* simulations. The analytical expression relates the approximate diffusion coefficient, D , and unbinding rate, k_{off} , needed to see depletion at a given length scale L and under a flow rate of v . The values $f_1 = 1$ and $f_2 = 0.333$ lead to an excellent fit between the analytic and simulation results for the 50% boundary. For *pombe* we set $L = 2 \mu m$ and

$v = 3 \times 10^{-3} \mu\text{m/s}$ as in Gerganova et al. and find that $D < 0.006 \mu\text{m}^2/\text{s}$ or $k_{\text{off}} < 0.005 \text{s}^{-1}$ result in depletion. To connect these two limiting values we assume a form for $D = (1 - \tau k_{\text{off}})/(\tau C)$ where τ and C are fixed by the limiting values. For *N. crassa* we set $L = 5 \mu\text{m}$ and determine $v = 0.1604 \mu\text{m/s}$ which is the expected flow rate due to exocytosis at an arclength of $1.589 \mu\text{m}$. For *N. crassa*, $D < 0.802 \mu\text{m}^2/\text{s}$ or $k_{\text{off}} < 0.107 \text{s}^{-1}$ leads to depletion. While the boundary shifts to larger D and k_{off} values for *N. crassa*, this analysis shows that a dissociation rate similar to that of Cdc42-GDP (0.17s^{-1}) would provide sufficient mobility to withstand the membrane flow. We have added this analysis to the manuscript.

Figure R9. Depletion boundaries for *S. pombe* and *N. crassa* determined by either particle-based simulation (lighter lines) or from the dimensional analysis given in Gerganova et al. Sci Adv 2021.

4. In the Methods section on how they did the averaging to get particle concentration profiles, they refer to the “largest cluster” of Cdc42, and to the changing calculation of where the centroid is. It would be good to know how many clusters of Cdc42 are produced, and with what distribution of sizes and lifetimes. Does the model usually have an obvious dominant cluster with rare satellites, or are multi-cluster states common? Also, how stable is the cluster centroid position? If there is usually an obvious dominant cluster, on what timescale does the centroid change position, and by how much?

To address this question, we measured the number of clusters and their sizes as a function of time using a cluster cutoff distance of $0.7 \mu\text{m}$ (Fig. R10). The cluster cutoff distance is the maximum arclength distance two Cdc42-GTP particles can be separated by and still be considered as part of the same cluster. Fig. R10 shows there is always a single large cluster after steady state has been reached for Cdc42-WT and Cdc42-1ritC. For Cdc42-2ritC, a more diffuse patch forms at steady state that transiently breaks up into separate patches and reforms. For Cdc42-3ritC there are several small zones of activation with no dominant patch region at steady state. Therefore, as Cdc42-GDP mobility slows, the number of clusters of active Cdc42 increases.

Figure R10. Cluster sizes over time for WT and ritC conditions, where color indicates the number of clusters at time t , for the simulations of Figures 1E, 3A and Movies S1-S4. We used a cluster cutoff of $0.7 \mu\text{m}$ (i.e. particles within an arclength of $0.7 \mu\text{m}$ from each other are considered to be in the same cluster) and ignored any clusters with fewer than five particles of Cdc42-GTP.

For the same cluster definition, we also calculated the root mean square displacement (RMSD) of the largest Cdc42-GTP patch centroid for the WT and ritC conditions (Fig. R11, left). This shows that the WT, 1ritC, and 2ritC patches remain constrained to a region of the full sphere surface throughout the simulation. This constraining effect appears due to the presence of sGAP since in WT simulations without sGAP the RMSD value increases more quickly (Fig. R11B). We note that the 3ritC patch fluctuates more wildly and has much higher values of RMSD, indicating hotspots changing position on the order of hundreds of seconds by around $2 \mu\text{m}$, though still remaining constrained to approximately half the sphere surface even after 10^4 s (Fig. R11). The exact value of these displacements depends on the cluster cutoff distance used, but the behavior described is independent of this choice.

Figure R11. RMSD, averaged over each starting time t_0 for each time lag τ , of the centroid of the largest Cdc42-GTP cluster for WT and XritC conditions. Averaging started after patch formed at least 5500 s after the initial random placement of particles on sphere surface. Panel A show reference case for WT and ritC conditions with sGAP. Panel B shows the WT case without sGAP and with k_{hydro} increased to 1.75 s^{-1} to allow a single, well-defined Cdc42-GTP patch to form (see Figure R8). Dashed blue line is the average distance between two random points on the surface of a sphere with radius $1.8 \mu\text{m}$.

5. There seemed to be contradictory statements on how Cdc42-GTP diffusion and detachment were modeled for the ritC constructs. In the Results it says “we assumed that the ...Cdc42-GTP...parameters are limited by the ritC anchor and set them the same as for Cdc42-GDP”. This would seem to imply that GTP- and GDP-bound forms now share the same mobility. But in the Methods it says they “lowered the corresponding values of Cdc42-GTP ... so that Cdc42-GDP would always have a greater than or equal mobility compared to Cdc42-GTP”. I don’t understand how both statements can be true. I would assume the Methods version is correct?

The Methods version is the correct statement. We apologize for the lack of clarity in the Results section version – the statement cited above was meant to just focus on the ritC constructs. For all three ritC constructs the diffusion coefficient for Cdc42-GDP, D_D , is lower than the diffusion coefficient for Cdc42-GTP in the WT case, $D_{T,WT}$, so that the expression in the Methods section will simplify to $D_T = \min\{D_D, D_{T,WT}\} = D_D$. The reasoning is similar concerning the unbinding rate for Cdc42-2ritC and Cdc42-3ritC, where r_D is lower than $r_{T,WT}$ so that $r_T = \min\{r_D, r_{T,WT}\} = r_D$. For Cdc42-1ritC the unbinding rate, r_D , is higher than $r_{T,WT}$ so that $r_T = \min\{r_D, r_{T,WT}\} = r_{T,WT}$ and so the GTP and GDP bound forms have unique unbinding rates in this case. We updated this sentence in the Results section to be clearer as follows:

“The two slower Cdc42-ritC constructs (Cdc42-2ritC and Cdc42-3ritC) have lower Cdc42-GDP diffusion and dissociation constants than WT Cdc42-GTP (Figures 1D, 2B, S3); thus, for the 2ritC and 3ritC cases we assumed that the corresponding mobility parameters of the GTP-bound form, in association or not with Scd1/2, are limited by the ritC anchor and set them the same as for the GDP-bound form.”

6. For the simulations of diploid cells, particle numbers were doubled. But was the sphere volume also doubled? If not, aren't these simulations of overexpressing cells rather than diploids?

We thank the reviewer for pointing out this discrepancy. Indeed, our prior “diploid” simulations had doubled particle numbers but were on the same sized sphere. We fixed this issue by doubling the volume as well as the particle number. The qualitative result that having a WT copy rescues patch formation remain unchanged and the distribution of Cdc42-XritC matches the experimentally observed ones with depletion of Cdc42-3ritC. We display these new results in the updated Supplemental Figure S5.

7. It was not clear which combinations of GAP deletions (other than the triple) were able to suppress the 3x ritC Cdc42 lethality. Do single GAP deletes suppress? If so, which ones? Do they look any better than the triple? Simulation of GAP deletion cells is described only for the sGAP: what happens if one lowers khydro instead?

We were also very interested in finding out whether other combinations of GAP deletions would suppress lethality of slow Cdc42. Our experiment was to sporulate heterozygous mutants for all GAP and Cdc42 loci, which could in principle yield any combination of mutants, and so we were hoping to find all the viable combinations. In this experiment, when genotyping the viable colonies obtained from germinated spores, we initially recovered other combinations than the triple GAP mutant. However, in further PCR tests to verify genotypes, the single/double mutants contained not only the *cdc42-3ritC* slow allele but also WT *cdc42* coding sequence, suggesting that they had arisen from aberrant meiotic figures. The triple mutants contained only the *cdc42-3ritC* slow allele. While this suggests that other combinations of GAP mutants do not support viability with the slow *cdc42-3ritC*, we prefer to remain careful with this negative result and abstain from making strong statements in the paper.

To be certain about the viable *cdc42-3ritC* triple GAP mutants, we also obtained them from tetrad dissections, and these are the alleles we used to show morphology in Figure 4C.

We also performed simulations to check the effect of reducing khydro in simulations. We find that reduction of khydro does lead to an increase in Cdc42-GTP together with a narrowing of the patch. However, for our reference WT parameters, this effect is weaker than that of reducing sGAP by 50%. It

is also not sufficient to bring active cdc42-3ritC cells above the survival threshold (dashed line below in Figure R12 and updated in the main Figure 4A).

Figure R13. Updated Figure 4A that includes the case where k_{hydro} is reduced to 1/100th of its reference value. This does increase the amount of Cdc42-GTP on the surface, but the effect is not as strong as reducing the amount of sGAP.

8. There is a typo in the Exo/endocytosis Methods where it says “wexo or wexo, respectively”. Presumably one of those is intended to be wendo.

Thanks - corrected!

REVIEWER COMMENTS

Reviewer #1 (Remarks to the Author):

We thank the authors for revising the manuscript thoroughly.

We acknowledge the experimental progress and that the authors invested in new numerical simulation on spherocylinders. However, we remark that we still have reservations concerning the theoretical model as it does not include local changes in the membrane area due to addition (exocytosis) and removal (endocytosis) of membrane material. We highlight that this refers to a localized change in area and does not correspond to the growing domain as implemented in the revised version. We expect that a flux-mediated description may only be valid for a single exo-/endocytosis event where a constant frame of reference is present.

Furthermore, we have the following minor requests:

- Several references lack author names [39,42,...], volume/page numbers[8,33,42,...] or include unspecified strings [18,55]. We kindly ask the authors to revise the reference list.
- The authors should state the necessary language standard needed to compile the provided code.
- We kindly ask the authors to add comments to the code, outlining its structure, and, in particular, explain the numbers provided within the `avgDisplacementAtArclength()` method. In addition, to improve the readability of the code, we ask to clean up the code.

We acknowledge the experimental significance and would, in light of the new simulations, recommend publishing the manuscript after implementing the requested changes above.

Reviewer #2 (Remarks to the Author):

In the revised manuscript, the authors have addressed the comments raised regarding the initial submission. The revised manuscript reads more clearly than the original submission, and I now recommend publication.

Reviewer #2 (Remarks on code availability):

The code provided is a great resource for the community.

Reviewer #3 (Remarks to the Author):

The revisions have improved the paper and taken care of all my concerns. I am happy to recommend publication of this interesting study.

We thank the reviewers for their careful reading of our work and their insightful comments. We have addressed them through substantial modelling work and some additional experiments. Detailed responses are provided below between individual reviewers' points.

Reviewer #1 (Remarks to the Author):

The authors examine the role of effective membrane flows induced by exo-/ and endocytosis on the stability of the formation of polar Cdc42 domains, vital for cell division in fission yeast. They combine experimental studies of yeast cells modulating membrane binding dynamics of Cdc42 (1ritC, 2ritC, and 3ritC) with a mesoscale particle-based reaction-diffusion model. Within the framework of the proposed model, the authors provide a rationale for the experimentally observed behavior.

The primary mechanism, namely that proteins with strong membrane binding affinity experience an effective outflux from sites with predominant exocytosis, has already been studied previously by the same group in [V. Gerganova, I. Lamas, D. Rutkowski et al., Sci. Adv. 7, eabg6718 (2021), manuscript ref. 22]. It is imperative that the authors explain the significant advances in their current work compared to this earlier study. Notably, the previous study already includes all the relevant mechanisms and even incorporates experiments with the Cdc42-xritC constructs like the ones presented in this manuscript.

In terms of modeling, the only novelty seems to be that the authors now included the apparent mechanism of detachment of proteins into the cytoplasm, where they assume infinite diffusivity. This may clarify some details of the initially presented mechanism, but in the absence of qualitative differences, we wonder if this represents a sufficiently significant novelty. Compared to their earlier study (Ref. 22), this manuscript appears to be only an incremental next step, which is noteworthy but not broad enough for the intended readership of Nature Communications.

The previous study (Gerganova et al, Sci Adv 2021) established the basis of in-plane membrane flows, showing that polarized secretion coupled to endocytosis over a broader area causes net membrane flow away for the site of secretion and that membrane-associated proteins with long enough residence time at the membrane couple to this flow and are displaced from sites of secretion. We did not address – neither through modelling, nor through experiments – how Cdc42 molecules themselves may be influenced by the flow. On the modelling side, we assumed a stable site of secretion and only looked at downstream consequences. On the experimental side, contrary to what the reviewer states, we did not perform any experiments where we modified Cdc42 membrane binding. Our constructs with 1/2/3-ritC involved either GFP or CIBN and were used as read-outs of membrane flows.

The advance here is to probe how Cdc42 molecules, whose activity establishes the flows, are themselves affected by membrane flows, i.e. we no longer simply consider flows as a downstream consequence of local Cdc42 activity, but as an element that feeds back to Cdc42, by both modifying the distribution of Cdc42 itself and its negative GAP regulators, and can thus modify its own source. In terms of modelling, this is a significant advance as it integrates Cdc42 symmetry-breaking feedback system (modeled as reacting particles) on a flowing biophysical support. Experimentally, we show that the system breaks down when Cdc42 mobility is decreased. This is a clear demonstration that the Cdc42 polarity system can only be maintained through turnover of Cdc42 that is faster than the underlying support.

We have now been more explicit on these advances in the introduction to the manuscript.

Another serious concern is that there is a major overlap between the present manuscript and Ref. 22: Several figures are reproductions (e.g., 1C, 1D, S1A, and S1B), some without any indication, and entire blocks of text are one-to-one copies. The novel results and differences to Ref. 22, if present, are unclear to us. There is also a significant lack of details in the description of the model and the numerical simulations of the latter. This renders reproducing the study almost impossible.

We are sorry for the confusion. This apparent overlap is due to our use of results from our prior work as the basis and motivation for our new work. We re-used a few illustrations from Gerganova et al, with modifications, as a starting point to introduce this new study.

The full model is described by Fig. 1A, 1B, 1C where only part 1C of the model is the same as in Gerganova et al., hence the use of this panel. In our technical description of the part of the model that corresponds to Fig. 1C, we had tried to condense a more detailed description in Gerganova et al. but we understand this came out as both cut-and-paste and lack of detail.

Fig. 1D shows our new measurements of Cdc42 mobilities in relationship to the depletion regions discovered in Gerganova et al.

We did not intend to present S1A and S1B as new results but rather added them to help the readers with some key background observations so that they don't have to extract them from Gerganova et al.

The above are now stated much more explicitly in the legends. We also contacted Science Advances to confirm we have permissions for re-use. We paste here the reply from the journal: "Per the Science Advances Author License to Publish, after publication of your Science Advances paper, no further permission is needed from AAAS in order for you to use/modify those figures & tables in a new work you write (eg: book chapter, journal article, dissertation, blog post etc.). This policy applies to your use in both non-commercial and commercial works, so long as you credit the original Science Advances paper and indicate the type of CC license it was published with."

We have added additional details in the description of the model and numerical simulations in response to specific reviewer comments (see below). As originally intended, we have added a Code availability section to the manuscript and the code is now available at <https://github.com/davidmrutkowski/Cdc42ParticleFlows>, allowing full reproducibility.

In addition to the above concerns, we have the following questions and requests:

- The reproduction of figures from other publications should be indicated.

This is now done in the legends.

- The authors introduce the concept of 'mobility' as a combination of membrane diffusion and detachment with subsequent fast cytoplasmic diffusion. These two contributing effects should be more clearly disentangled.

We understand the reviewer's concern and we have clarified in the text where there may be confusion. We still find it convenient to use a single term for brevity purposes. However, we hope it is clear that we did try to characterize and resolve the separate contributions of membrane diffusion and membrane dissociation/association. There is a difference between these two mobility modes as diffusion tends to broaden concentration peaks on the membrane over time while dissociation

followed by fast cytoplasmic diffusion and re-association leads to a gradual reduction of a membrane concentration peak height and increase in the surrounding values outside the peak, without broadening of the peak. Simulations also show that these two mobility mechanisms are not entirely equivalent (e.g. Figure 3C where the system behavior is generally different along the two bands). However, the difficulty in resolving these two modes of protein mobility is suggested by our fits to FRAP results, which generally have two best-fit bands at a finite koff / zero diffusion coefficient and at a finite diffusion coefficient / zero koff (e.g. Gerganova et al, Sci Adv 2021, Fig S4). To address this issue and trim down these bands as much as possible, we averaged fits to FRAP for bleaches of different widths, which have different sensitivity to detachment and diffusion.

- It is unclear whether the proposed shifting of agents after exo-/endocytosis events captures the correct behavior of the system as one neglects volume and surface extension and the resulting dilution. It is necessary to properly include patch generation and dissociation by endo-/exocytosis in the model.

We acknowledge that our simulations capture only a steady-state situation and not the full events of growth. We note that the timescales of surface and volume growth, which lead to cell shape changes, are much longer than the timescales of the molecular reactions we simulate, which provides support for our approach. We also note that membrane remodelling (i.e. local surface extension) is not neglected, as molecules are displaced as consequence of individual exo-/endocytic events.

However, to address the reviewer's concern, and to further explore the assumptions of our model, we created new simulations that include volume and surface expansion. In fission yeast, the spherocylindrical geometry is critical in accounting for volume expansion through cell elongation, therefore we implemented particle-based reactions along a growing cell tip. One major difficulty in producing simulations that reflect realistic growth in volume and surface is that growth is a consequence of Cdc42 local activity. Thus, the shape of the simulated surface/volume should in principle evolve as a consequence of the formation of the Cdc42-GTP patch. This poses substantial difficulties, as no element of our simulations ensures that zones of Cdc42 activity should coincide with the tip of a simulated spherocylindrical shape. Indeed, previous work has shown that the shape of the cell may by itself cause instability in the position of the Cdc42 patch (Trogon et al, PLoS Comp Biol 2018), which can be stabilized through mechanical feedback (Banavar et al, PLoS Comp Biol 2021). Thus, implementing growth on a self-polarizing system is not trivial.

To simplify the problem, rather than implementing membrane flows based on discrete events for endocytosis and exocytosis defined by the position of Cdc42-GTP, we used an average, azimuthally-symmetric profile of membrane flow centered at the cell tip. Working in the reference frame where the cell tip is at rest, in these simulations each particle is displaced radially along the spherical cap or cylindrical segment according to an average velocity that accounts for cell growth and net membrane area added or removed. This velocity flow profile, shown in Figure R1, was determined by: (1) Freezing Cdc42-GTP particles in a patch with width equal to the average patch from steady state WT/1ritC simulation on the sphere surface as in Figure 1E and 3A. (2) Running the simulation with endocytosis and exocytosis events generated by the stationary Cdc42-GTP and calculating the induced flow on probe ghost particles placed uniformly on the sphere. The ghost particles were used to accumulate the average flow and were not displaced during these simulations.

Figure R1. Average flow velocity away from the north pole due to random exo/endocytosis events under growth conditions occurring for the WT (red) and 1ritC (blue) distribution of Cdc42-GTP

The spherocylinder simulations begin with the same surface density and number of particles as our sphere simulations. We initialize the cylinder length to be $3.0 \mu\text{m}$ with the spherical radius remaining at $1.8 \mu\text{m}$ as in all other simulations. For each timestep we determine the displacement on the spherocylinder surface due to the flow velocity in Figure R1 and diffusion. We first displace the particle on the surface it begins on (e.g. hemisphere or cylinder) and then if the particle would cross the boundary between two surfaces, we advance the particle up to this boundary and continue the displacement on the other surface. Reflecting boundary conditions were applied at the free edge of the cylindrical segment. We performed simulations where the reflecting boundary was moved at a rate equal to the rate of tip growth, or else kept stationary. Since cell growth is relatively slow compared to the time of steady patch formation, we kept the number of particles fixed throughout the simulation (Figure R2). To compare the spherocylinder geometry results, both with or without explicit growth, to results using spherical geometry, we implemented the same algorithm for a sphere, assuming the flow originates at a fixed location on the sphere.

Figure R2. Spherocylinder growth simulations where cylinder surface ends in a reflecting boundary (WT case). The shifting downwards of this boundary is associated with net cell growth (net elongation rate of $0.031 \mu\text{m}/\text{min}$). All sGAP particles are initiated in the cylinder portion of the surface to avoid transient patch formation outside the tip region.

We find that all three cases (growing spherocylinder, static spherocylinder, and sphere) lead to a steady state patch of similar width and concentration amplitude to each other (Figure R3A), as well as to the simulations of Figure 1E and 3A. Because we initialized the sGAP outside the top hemisphere, we observe the same lateral accumulation in sGAP concentration at $\sim 3.5 \mu\text{m}$, as seen in Fig. 1E, at much later times (Figure R3B). For these later times, the Cdc42 patch weakens in the case with spherocylinder growth due to growth-induced particle dilution (Figure R3B).

The observation that different geometries do not influence the size of the patch in these simulations demonstrates that our sphere simulations can be used as a good approximation for polarization of a growing spherocylinder. We added these three figures as new Supplemental Figure 2 and added a few sentences in the main text and a subsection in the Modeling methods section describing these spherocylinder simulations.

Figure R3. Concentration profiles comparing polarization under WT and 1ritC conditions for growing spherocylinder, static spherocylinder, and sphere, using particle displacements due to endocytosis and exocytosis as in Fig. R1. Simulations were run for 5000 s. (A) Concentration profile averages at intermediate times after patch formation (400 to 1000 s), as indicated in the panels. (B) Concentration profile averages at longer times (2500 to 5000 s), as indicated in the panels. A concentration peak of sGAP at ~ 3 - $3.5 \mu\text{m}$ is observed at long times due to flow displacement. The Cdc42 patch weakens in the case with spherocylinder growth due to growth-induced particle dilution.

- The model description needs more clarification in the main text and the Methods section. In particular, equations for the reaction kinetics and the functional dependence of the displacements $\Delta r(s)$ should be given.

We have added the model reactions in a table (new Table S1) and the functional dependence of the displacement $\Delta r_{exo/endo}(s)$ to the Mathematical Modeling Methods section. We list the model reactions in a table, following earlier work using particle simulations [Pablo, Ramirez, Elston *PLOS Comp Biol* 2018]. Mathematically representing our discrete particle reactions in the form of equations would involve writing equations in a field operator representation [Doi *J Phys A* 1976 9:1465, 9:1479], which we find to be unnecessarily complex for the purposes of this work. Since our simulations are not performed at the continuum level, our model does not reduce to partial differential equations such as

$$\frac{\partial Cdc42T}{\partial t} = (\lambda_{SD}Scd + \lambda_{ST}ScdCdc42T) \cdot Cdc42D - k_{hydro}Cdc42T - r_T Cdc42T - \lambda_{GAP}Cdc42T \cdot sGAP - \lambda_{ST}Scd \cdot Cdc42T + r_{ST}ScdCdc42T + \nabla(\mathbf{j}_{Cdc42T}),$$

where the last term represents the flux of the species on the membrane due to the diffusion and flow due to exocytosis / endocytosis events.

- *We'd like to ask the authors to discuss in more detail the effect that gives rise to the effective membrane flux, namely the overall predominant increase in membrane area due to exocytosis. To some extent, this is obscured by the different values for endo-/exocytosis areas and corresponding rates.*

The effective membrane flux away from the tip is driven in part by the difference in net area addition/removal and in part by the wider profile of endocytosis compared to exocytosis. Indeed, in Gerganova et al. we showed that having neither of these aspects ($\sigma_{\text{exo}} = \sigma_{\text{endo}}$ and no net growth) leads to no strong depletion of inert particles from the tip region (Fig. S2C in that work). Both aspects contribute additively to the overall displacement. In the model presented in this work, there is an overall net addition of membrane since exocytosis delivers $0.0214 \mu\text{m}^2/\text{s}$ while endocytosis removes $0.0157 \mu\text{m}^2/\text{s}$. This difference derives from the fact that we are modeling a growing tip of *S. pombe* which grows at a rate of $20.7 \mu\text{m}^2/\text{h}$ as outlined in Gerganova et al. We modified parts of the first section of the Results section to make these points clearer as follows: "Membrane flux is due in part to the width of the exocytosis profile being narrower than that of endocytosis (Gerganova et al. Sci Adv 2021), and in part to the higher rate of exocytosis than endocytosis ($r_{\text{exo}}A_{\text{exo}} = 0.0214 \mu\text{m}^2/\text{s}$ while $r_{\text{endo}}A_{\text{endo}} = 0.0157 \mu\text{m}^2/\text{s}$) leading to net cell growth. For simplicity we report results for flows on the surface of a sphere without explicitly simulating growth, assuming the center of a Cdc42 patch corresponds to the frame of reference of a growing tip".

- *The authors should describe more precisely how cytoplasmic-membrane particle reactions are implemented.*

Unimolecular cytoplasmic-membrane particle reactions are implemented as described in the *Particle reactions* sub-section of the Methods section where a particle in the cytoplasm is placed anywhere on the sphere surface (fast cytoplasmic diffusion assumption) with equal probability with a rate according to its binding rate as given in Table 1. Bimolecular cytoplasmic-membrane particle reaction only involves cytoplasmic Scd1/2 reacting with membrane bound Cdc42-GTP. We have added the following sentence to the *Mathematical Modeling Methods* section to describe how the final position of this resulting complex is determined: "For the reaction between cytoplasmic Scd1/2 and membrane-bound Cdc42-GTP we place the resulting complex at the position the Cdc42-GTP had since it is the only particle on the membrane for this reaction. Dissociation from the membrane places the particles in the
cytoplasmic
pool."

- *The authors should show equations corresponding to Figure 1A.*

As mentioned above, we represented the reactions as a list in new Table S1.

- *We ask the authors to provide more details on the numerical simulations and access to the code to make the study reproducible.*

We have added additional details in the description of the model and numerical simulations. The code is now available at <https://github.com/davidmrukowski/Cdc42ParticleFlows>.

- How is the time step in the Gillespie algorithm related to the dynamics in the Brownian dynamics simulation?

All reactions are performed with the same discrete timestep as for the Brownian diffusion dynamics ($\Delta t = 0.001$ s), except for exocytosis/endocytosis and unimolecular membrane binding reactions that were based on a varying timestep determined by the Gillespie algorithm, independent of this discrete timestep. The timesteps determined from the Gillespie algorithm occur in between discrete timesteps as shown in the timeline in Figure R4. We use the Gillespie algorithm for exo/endocytosis events that occur less frequently. Since membrane binding reaction rates do not depend on spatial position, and could also occur rarely (such as for Cdc42-xritC3), we also used a Gillespie algorithm for these reactions. As a result, for WT cells, both the Gillespie and discrete timesteps are typically around the same value. Diffusion for every particle in the system occurs to advance the system up to discrete timesteps and to times associated with exo/endocytosis. To handle the split between Gillespie and discrete timestep, each particle has two times associated with it; t_{react} , which is the most recent time the particle was allowed to react up to, and t_{pos} which gives the most recent time the particle was diffused up to. Specifically, the probability to react is determined by $P_{react} = 1 - \exp[-r(n\Delta t - t_{react})]$, where n is the number of discrete time steps. Particles are diffused over the time period of $t - t_{pos}$, where t is the current simulation time. Upon membrane binding of a specific particle, both t_{react} and t_{pos} are set equal to the current simulation time. We have added these clarifications in the *Time Step* section of the *Mathematical Modeling Methods* section.

Figure R4. Timeline showing placement of discrete timestep events ($n\Delta t$) and events that occur at times determined using the Gillespie algorithm (dashed lines).

- We ask the authors to comment on the implied dependence of the membrane binding rate of Cdc42-GDP on the dissociation rate constants of Cdc42-xritC-GTP/-GDP stated on page 11.

We have clarified this equation in the main text in the *Mathematical Modeling Methods* section (*Assumptions of particle reaction scheme* section):

“The membrane binding rate of Cdc42-GDP, k_D , was set dependent on the dissociation rates r_T , r_D of Cdc42 GTP/GDP so that approximately 3/5 of Cdc42 is on the membrane. We determined the functional form of this dependence from the steady state balance equation $k_D N_{cyto} = N_T r_T + N_D r_D$, where N_{cyto} , N_T , N_D are the number of Cdc42 in the cytoplasm, and bound to the membrane with GTP or GDP, respectively. Additionally, we fix $N_D = 2\beta N_{cyto}$ and $N_{cyto}/(N_{cyto} + N_D + N_T) = 1/(1 + 3\beta)$, where for our simulations $\beta = 0.513$ in order to have a ratio of 3/5 of the total Cdc42 on the membrane. Solving these three equations leads to the form shown in Table S2: $k_D = \beta(r_T + 2r_D)$. Indeed, for the reference WT-Cdc42 simulations $N_D \approx 2200$, $N_{cyto} \approx 2220$, $N_T \approx 380$, and $N_{scd12T}^{WT} \approx 200$ ”.

We note that this calculation of k_D assumes that the binding affinity of Cdc42-xritC constructs for the plasma membrane remains approximately the same as for WT cells. We checked that our results do not depend on this assumption as shown below in Figure R5 (also provided as Figure S10 in the

manuscript). The main difference is an overall higher level of Cdc42 on the membrane when using a constant value of $k_D = 0.1 \text{ s}^{-1}$ for all three Cdc42-xritC constructs (compare Figure R5A to R5B).

Figure R5 (A) Profiles for ritC conditions at steady state taken from Figure 3A (B) Profiles from simulations where the membrane binding rate of Cdc42-GDP, k_D , was fixed at a value of 0.1 s^{-1} .

- The authors mainly describe the observed behavior for the stability of the Cdc42-ATP polar domain and the GAP depletion experiments. Additionally, a mechanistic explanation based on the proposed model would be desirable.

We have modified our Discussion section slightly to focus more on how the model generates patches both with the discrete GAP molecule and without it as shown below:

“Our findings, supported by experimental validation, demonstrate that differential dynamics of fast Cdc42 and slow inhibitor provide a mechanism for pattern formation in the presence of membrane flow. In this mechanism, membrane flows enable protein dispersion that both constrains the membrane affinity of polarity factors and allows long-range displacement of inhibitors, implementing a transport-driven lateral inhibition mechanism as in classic activator-inhibitor pattern formation. While prior studies showed that Cdc42-GTP relative low mobility traps it within its activation region, our study, as discussed below, reveals that Cdc42 diffusion coefficient on the membrane, and membrane dissociation rate, should nevertheless be fast enough for the average duration of Cdc42 within the active zone to be shorter than the characteristic time for membrane flow to displace it, in order to generate a zone with clearly defined boundaries.

[...] While membrane flows promote polarization through removal of negative factors, they also destabilize the polarity patch, which replenishes through rapid Cdc42 flux, in agreement with previous modelling effort. We now show, both in silico and experimentally, that sufficiently fast dynamics (fast membrane unbinding rates and/or fast lateral diffusion) of Cdc42 is required for efficient polarization. The loss of polarization upon decreased Cdc42 mobility can be understood in the following way: Arrival of secretory vesicles at the patch causes local displacement and dilution of Cdc42, leading to widening and weakening of the patch. Consequently, secretory vesicle delivery is less polarized, reducing the

net membrane flow. sGAP is thus less efficiently displaced, causing stronger overlap between sGAP and Cdc42-GTP and consequent Cdc42 inactivation. By contrast, fast dynamics allow Cdc42 to rapidly repopulate the zones of secretion, alleviating the effects of the membrane turnover.[...] We conclude that establishment and maintenance of a polarized patch requires its components to turn over faster than the surface on which it is formed.

In principle, a high Cdc42-GTP turnover at the membrane can result not only from fast Cdc42 protein mobility, but also from a high rate of GTP hydrolysis. Thus, an alternate mechanism for patch formation in the presence of flows that displace Cdc42, but in absence of sGAPs, may involve a higher level of k_{hydro} (equivalent to Rga3 overexpression compared to our reference case, and provided that Cdc42-GDP mobility remains sufficiently high)."

- Throughout the introduction paragraph, various statements lack proper reference. For instance:
 - o 'Polarized secretion, the directional and selective release of molecules from a cell, is critical in many biological processes, including cell communication, tissue development, and immune response. In fungi and other walled cells, it underlies polarized cell growth, as it is necessary for the local delivery of cell wall remodeling and membrane material. Yeast model organisms [...]',
 - o 'The Cdc42 polarity system in yeast is believed to represent one of the best examples of biological pattern formation on cell membranes [...]',
 - o 'Cdc42 accumulation is thought to be driven by the slower mobility of Cdc42-GTP than Cdc42-GDP at the plasma membrane. '
 - o The whole paragraph 'Prior modeling work ...' on p. 3

We have added references where they were missing in these sections.

In summary, we have strong reservations regarding the originality and potential broad interest of this work.

We hope we have satisfactorily explained the advance of this study. In terms of originality and broad interest, this is the first study that examines in detail how membrane flows induced by polarized secretion modulate the establishment of the polarity domain. The concept that the proteins that define the polarity patch need to exhibit faster mobility than the membrane support on which they assemble is a broadly valid one, with direct relevance to understand polarized cell growth in any organism.

Reviewer #2 (Remarks to the Author):

*This research article presents a theoretical and experimental study of how cell polarization regulated by Cdc42 influences zones of polarized secretion and associated membrane flows. Spontaneous symmetry breaking during the polarization of yeast (*Saccharomyces cerevisiae* and *Schizosaccharomyces pombe*) has been studied extensively in the past and is a topic of considerable interest and debate. The noteworthy result here is the clear demonstration that changes in *cdc42* mobility directly affect polarized secretion. This is an interesting contribution that is well-written and ought to be published after the authors have considered my suggestions.*

I would like the authors to address the following points.

1. The authors have previous work on membrane flows, notably cell patterning by secretion-induced plasma membrane flows. The paper would be strengthened by clearly addressing the differences/additions to their and other authors' previous findings like in Jose M et al. J Cell Biol, 2013. Their new finding pertains to how secretion is affected by a modulation of the mobility of Cdc42. A detailed discussion of the model's assumptions and its relationship to prior work is needed to clarify the novelty of the proposed mathematical model with published research. This paper should, at a minimum, cite and discuss the following papers looking at membrane flows during yeast polarization:

There are two major differences between our current work and earlier work, including our previous Gerganova et al, Sci Adv 2021 and the Jose et al, JCB 2013 paper cited above. First, we explicitly model the effect of membrane flows – i.e the remodeling of the plasma membrane by exocytic and endocytic events – on the displacement of Cdc42 proteins, and probe the importance of Cdc42 mobility at the membrane. Second, we explicitly model the effect of Cdc42 GAP proteins, which are themselves displaced from the polarity site by the flow. Our work comes to the conclusions that Cdc42 needs to turn over faster than the rate of membrane flow and that the flow-mediated GAP displacement promotes the formation of a polarized patch.

Most prior modelling efforts did not take membrane flows into account: Jose et al, JCB 2013 models how endocytosis can occur around zones of exocytosis, where polarity factors are simply dispersed from sites of delivery at secretion sites through diffusive motion; Gerganova et al, Sci Adv 2021, modelled the effect of membrane flows on the distribution of membrane-associated proteins, but assumed a static site of secretion itself unaffected by flows. To our knowledge, only two prior models explored the effect of polarized secretion on cell polarity establishment by simulating new membrane insertion at the plasma membrane: 1) Layton et al, Curr Biol 2011, concluded that polarized secretion perturbs cell polarity by diluting the Cdc42 patch even upon delivery of Cdc42 by secretory vesicles; 2) Okada et al, Dev Cell 2013 modelled a growing surface displacing septins from the site of secretion defined by Cdc42-GTP, leading to the formation of a ring of septins, which were proposed to recruit Cdc42 GAPs to feedback and restrict Cdc42 activity and growth of the bud in its center. This second study used a continuum concentration approximation rather than a particle-based method, and did not introduce endocytosis nor the concept of lateral membrane flow. Our results agree with these previous conclusions (weakening of the polarity patch by secretion, and exclusion of slow-mobile proteins such as septin filaments), which we also extend by showing a direct regulation through exclusion of GAP proteins and probing the importance of Cdc42 fast mobility. All these papers are cited in our manuscript.

2. As the cell tries to establish a single polarization cap from multiple potential polarization sites, which fail to progress, it grows isotropically. How does this isotropic growth (and the presence of multiple polarization sites and potential interaction between them) impact the membrane flows?

The reviewer asks an interesting question regarding the kinetics of approach to steady state in our model, and the impact of membrane flow. Quantifying these kinetics is somewhat complex and so we feel it goes beyond the scope of this work. We can however highlight some key aspects of this behavior. Indeed, during symmetry breaking in the model, multiple polarization sites can initially form, depending on Cdc42 mobility parameters (please also see Figure R10 in response to reviewer 3).

The evolution of these patches depends on the following dynamical mechanisms:

(1) There is competition between patches due to both protein mobility in/out of patches and the Cdc42 scaffold mediated positive feedback, and smaller patches are expected to weaken against larger ones, similar to reaction-diffusion models without membrane flows.

(2) Patches compete for membrane delivery that displaces sGAP. Since sGAP will infiltrate small patches if there is not enough membrane delivery, this mechanism leads to an enhanced rate at which small patches disappear, leading to the growth of larger patches.

(3) Membrane flows could allow two nearby patches to merge due to their combined membrane flow. The net membrane flow could displace the sGAP from the region between the two patches and generate a region that is devoid of sGAP, allowing the two patches to merge. To investigate this mechanism, we calculated the average membrane flows due to two patches of Cdc42-GTP, as shown below in Fig. R6. The membrane flow between the two patches can be seen to drive particles (that are not exactly along the centerline of the two patches) out of the region between the two patches.

Figure R6. Net membrane flow vectors on sphere surface for two patches of Cdc42-GTP (orange). The flows were calculated by placing static Cdc42-GTP particles (orange) and calculating the induced flow on probe ghost particles on the sphere (green arrows).

For WT simulations we did not observe the appearance of two competing patches, so positive feedback and membrane-flow displacement work simultaneously to establish a single patch. For conditions corresponding to Cdc42-1ritC cells, two clearly distinct patches do form (Figure R10 in response to reviewer 3 below). For this latter case we observe one patch growing at the expense of a shrinking one, without the two moving closer together, indicating that patch competition for Scd1/2 or membrane delivery (mechanisms 1 and 2 above) dominate over membrane-flow-induced lateral merging (mechanism 3) in the process of single patch formation.

We note that the process of patch emergence in the presence of flows cannot in principle be reduced to results of prior works that rely on non-linear positive feedback. In fact, reducing the strength of positive feedback by reducing λ_{STT} in the model leads to a reduction in the overall patch size and strength, but it does not abolish polarization into a single patch (Figure R7 and new supplemental Figure S8). This occurs for both WT and Cdc42-1ritC parameters even for $\lambda_{STT}=0$. For the Cdc42-1ritC case, a single patch emerges directly without an intermediate two-patch competing regime (not shown). We have added a paragraph in the Discussion section that describes this result.

Figure R7. Concentration profiles when reducing non-linear positive term, λ_{STT} , from reference value of 5 s^{-1} , keeping all other parameters same as WT cells.

3. The geometry for the simulations seems to be spherical. Have the authors tried to change the geometry to understand membrane flows? How do geometry and the cell wall affect the membrane flows? There have been recent theoretical models (Trogon M. et al. Plos CB 2018 and Banavar SP. et al. Plos CB 2021) looking at how geometry affects polarization during growth. These studies along with possible effects of the geometry and cell wall growth should be discussed in more detail.

These are interesting questions. We chose the sphere as this represents the shape of the cell pole of the *S. pombe* cells quite faithfully, where, to the best of our knowledge, all exocytic activity occurs. In response to this comment and also to Reviewer 1, we have simulated an azimuthally averaged exo/endocytosis version of our model on the surface of a growing spherocylinder. We kindly refer the Reviewer to our response above. Briefly, we found that essentially identical profiles develop on the surface of the sphere or the spherocylinder (Fig. R3 and new Supplementary Figure S2), which validates our use of the sphere.

The cited studies showed that some models of Cdc42 symmetry breaking were unstable in non-spherical shapes, notably in shapes resembling that of mating cells, i.e. cells with short growth projections. The second of these papers showed that stability could be obtained in these shapes through mechanical feedback between cell wall strain (signaling through Rho GTPase) and the Cdc42 polarization model. Investigating our full model integrating membrane flows in different morphologies is beyond the current study, but it would be interesting to test if the stabilization of the polarity patch by the local depletion of GAPs may also suffice (or contribute) to stabilizing the patch at the tip of cells with short projections. For instance, it is interesting that *S. cerevisiae* mating projection localize septins around their base, which (as proposed in Okada et al, Dev Cell 2013) may be displaced by flows and recruit Cdc42 GAPs. We have added the perspective to probe the role of membrane flows in the discussion.

4. The authors state: “The spatial probability distributions for exocytosis and endocytosis are convolutions of Cdc42-GTP particle positions with a Gaussian of width w_{exo} and w_{endo} , which we

calculated by comparing tip distributions of CRIB-GFP (marking Cdc42-GTP) to profiles of exocytosis and endocytosis in wild type cells.” Did the authors map the sites of exocytosis and endocytosis by looking at molecules involved in the exocyst complex such as Exo70 and WASp, implicated in clathrin mediated endocytosis?

We did that in Gerganova et al, Sci Adv 2021 (in Figure S3). The convolutions used here result in distributions that match the experimental profiles (Figure S4 of the revised manuscript).

Reviewer #3 (Remarks to the Author):

This paper extends the findings from the authors’ recent work (Gerganova et al) demonstrating that membrane flow clears slow-mobility membrane proteins from the polarity site in fission yeast. Whereas the previous work developed a modeling strategy for membrane flows but did not model polarity per se, this paper presents a combination of a Cdc42 polarity model (Pablo et al.) with the vesicle traffic/slow GAP model (Gerganova et al). Overall, this model seems well executed, and allows the authors to make some non-obvious predictions that are then confirmed by experiment. The main prediction is that slowing Cdc42 mobility can allow membrane flows to perturb and eventually destroy the polarized Cdc42 distribution. In general, one expects slowing the mobility of a polarity factor to make it easier to retain a polarized distribution of that factor. The point that coupling polarity to membrane flows can reverse this intuition seems important and well validated for the parameter space they explored, and the fission yeast situation.

Overall, I was persuaded by the authors’ arguments. But I was confused by some issues listed below. The first in particular seems very important to clarify: I am not persuaded that the FRAP fit approach they use is valid for the tip FRAP, or that a model parameter from that fit makes any sense.

1. I am confused by the reaction modeled with parameter rT . This appears to represent detachment of Cdc42-GTP from the membrane combined with GTP hydrolysis to yield cytoplasmic Cdc42-GDP. No such reaction was present in Pablo et al. While there is prior evidence for membrane detachment of Cdc42-GTP (which should be cited if the reaction is retained), that is thought to be much slower than membrane detachment of Cdc42-GDP (especially because GDI preferentially binds Cdc42-GDP). Yet here it occurs at the same rate. Why is this reaction present and how does its magnitude affect model performance?

The parameter rT is said to be estimated by fitting FRAP recovery of fluorescent Cdc42 at the cell tip, and it was my understanding that the fitted model includes only diffusion and detachment of Cdc42-GTP, with no consideration that Cdc42 detachment may occur through sequential GTP hydrolysis and detachment of Cdc42-GDP. Is that correct? If so, would the FRAP be equally well fitted by adjusting r_{hydro} and rD instead of including rT ?

To me, it would make sense to remove the rT reaction, unless there is independent support for that reaction. If this does not affect the main conclusions, then removing the reaction seems prudent. (Fig. 1F would seem to indicate that a rapid rT is partially responsible for the loss of polarity upon sGAP removal, but the overall conclusion that without sGAP polarity is mostly lost would remain.) If the rT reaction does materially affect the main conclusions, then I would like to see some support for that reaction (and its magnitude) presented and justified.

Thank you for this important point. The membrane detachment rate for Cdc42-GTP (rT) we considered was indeed an effective rT , estimated from the FRAP fits, but representing an aggregate between rates

of Cdc42-GTP hydrolysis and detachment of Cdc42-GDP (and perhaps Cdc42-GTP). The reviewer is also correct in that the main conclusions of the model generally remain valid without rT (rT = 0).

Nevertheless, Cdc42-GTP is not irreversibly stuck at the plasma membrane and previous work has shown PM-cytosol exchange of Cdc42-GTP, as also pointed out by the reviewer (notably Woods et al, Curr Biol 2016), which argues against complete removal of the rT reaction. We agree that FRAP fits at cell tips are more difficult than fits from FRAP at cell sides. To estimate a rate for Cdc42-GTP detachment from the membrane, independently of GTP hydrolysis, we have conducted additional FRAP experiments with a GTP-locked allele of Cdc42 (Cdc42-Q61L-mCherry^{SW}) expressed in addition to endogenous Cdc42. As Cdc42-Q61L-mCherry^{SW} localizes ubiquitously at the plasma membrane, we could perform FRAP at the cell side in regions of various sizes and use these to distinguish lateral diffusion from membrane detachment. For these experiments, we kept Cdc42-Q61L expression moderate and only analyzed cells that retained the normal rod morphology. The best fit to this FRAP recovery indicates a much lower, but non-null, value of the dissociation rate for Cdc42-GTP to that of Cdc42-GDP (see updated Figure 1D and S3C). The diffusion coefficient of Cdc42-GTP is not distinguishable from that of Cdc42-GDP within our resolution. This came as a surprise to us, as analysis of half-tip FRAP in our previous work (Bendezu et al, PLoS Biol 2015) had experimentally suggested that the diffusion coefficient of Cdc42 at cell tips, which should be dominated by Cdc42-GTP, is slower by roughly an order of magnitude. In that earlier work we did not have an estimate of the Cdc42-GTP dissociation rate, but we had nevertheless interpreted the fact that Cdc42-GTP is less mobile than Cdc42-GDP by roughly one order of magnitude to be due to slower lateral diffusion of Cdc42-GTP. While the scatter in D and k_{off} fit values in our current fits is significant, overall, our fits on Cdc42-Q61L-mCherry^{SW} FRAPs now indicate that the main difference between Cdc42-GDP and Cdc42-GTP mobility resides in slower detachment rates of Cdc42-GTP. Our former interpretation was partly based on the observation that deletion of the GDI *rdi1Δ*, which is the only characterized mechanism of Cdc42-GDP membrane detachment, causes only modest changes in Cdc42 mobility. The data we now show suggest that there are other modes of Cdc42 membrane detachment, which act preferentially on Cdc42-GDP (in agreement with observations in Woods et al, Curr Biol 2016). We re-ran all simulations with this new reference value of D and k_{off} for Cdc42-GTP where required and have updated Figures 1, 3, 4A and Supplemental Figures 4 and 5 with these new simulations. All simulation results were robust to these changes.

*2. The first paragraph of the Discussion says that “our findings...demonstrate that...differential dynamics of fast Cdc42 and slow inhibitor are critical for pattern formation”. This does not seem quite right to me. They did not examine whether increasing k_{hydro} (analogous to *Rga3* overexpression, adding fast GAP/inhibitor) would compensate for the loss of slow GAP: why wouldn't this fast-Cdc42/fast-inhibitor scenario work, even with some membrane flow? At minimum, such simulations would be needed before reaching the conclusion that differential dynamics of this specific type are critical.*

We thank the reviewer for pointing out this issue and agree that our formulation was not optimal. Our data show that the timescale of Cdc42-GTP at the membrane has to be shorter than the characteristic membrane flow to sustain a stable polarity patch. Our data further show that cells can tolerate a substantial decrease in Cdc42 mobility (a substantial increase in Cdc42 membrane residence time) because of the action of slow GAP, which counteracts the reach of its active form, together with positive feedback reactions.

To address the question whether an increase in Cdc42 hydrolysis rate would similarly extend the cell's tolerance to reducing Cdc42 mobility, we varied the value of k_{hydro} in simulations without sGAP (updated Figure 1F and additional cases of Figure R8, also shown in a new supplemental Figure S7

discussed in the Discussion section). There is a range of k_{hydro} values where a distinct Cdc42-GTP patch forms in simulations without sGAP but with a higher value of k_{hydro} than in our reference WT case ($= 0.175 s^{-1}$). At high values of k_{hydro} ($\geq 35 s^{-1}$), the patch becomes unstable because deactivation of Cdc42-GTP occurs too quickly. At low values of k_{hydro} ($\leq 0.0175 s^{-1}$) Cdc42 remains active too long and spreads nearly over the entire sphere surface. This result further supports the finding that the lifetime of a Cdc42-GTP molecule at the membrane needs to be short enough relative to membrane flow for a patch to form. We have changed the quoted statement to the following:

“Our findings, supported by experimental validation, demonstrate that differential dynamics of fast Cdc42 and slow inhibitor provide a mechanism for pattern formation in the presence of membrane flow.”

Figure R8. Simulations at WT parameters without sGAP, as a function of k_{hydro} . The reference value of $0.175 s^{-1}$ and a case with $0.875 s^{-1}$ are shown in Figure 1F. Increase of k_{hydro} by a few times compared to the reference value recovers a more focused patch. Increasing or decreasing k_{hydro} too much results in loss of Cdc42-GTP polarization.

3. Later in the Discussion they conclude that “establishment of a polarized patch requires its components to turnover faster than the surface on which it is formed”. This is an intuitively appealing formulation consistent with their data and model. But it does raise the question of how hyphal fungi like *Neurospora crassa*, which have dramatically faster membrane turnover at the tip than *S. pombe*, can manage to retain polarity sites. Must they have turnover of polarity components at rates that are orders of magnitude faster? Is that realistic? I would be curious to know how the authors think that this might be achieved. (A similar problem was pointed out by Savage et al 2012).

This is an interesting question. To address it, we used estimates of membrane turnover measured for *N. crassa* (Bartnicki-Garcia et al, Fungal Genetics and Biology 2018), which estimate new PM added by exocytosis about 150-fold higher and PM removed by endocytosis about 27-fold higher than in *S. pombe*.

We have calculated the corresponding 50% tip depletion boundaries for the case of 150-fold faster exocytosis and 27-fold higher endocytosis using both our analytical expression $f_1 D/v < L < f_2 v/k_{off}$ (light red) and simulations of passive particles (red line) as in Gerganova et al. (Sci Adv 2021) in Fig R9 below (and new supplemental Figure S9). Here f_1 and f_2 are numerical prefactors of order unity. The corresponding boundaries for *S. pombe* are in blue. For these passive particle simulations, the values of α and γ were both set to 1.0. For the *N. crassa* particle simulations we used a sphere with radius 5 μm , where exocytosis and endocytosis profile standard deviations were multiplied by 2.78 that for *S. pombe*. Additionally, exocytosis and endocytosis rates were multiplied by 150 and 27 that for *S. pombe*, respectively, in order to match to the total rate of membrane addition/removal as given in [Bartnicki-Garcia et al, 2018]. Vesicle size was assumed to be the same as in Table S3. The exo/endocytosis cutoff was also increased to 5 μm for the *N. crassa* simulations. The analytical expression relates the approximate diffusion coefficient, D , and unbinding rate, k_{off} , needed to see depletion at a given length scale L and under a flow rate of v . The values $f_1 = 1$ and $f_2 = 0.333$ lead to an excellent fit between the analytic and simulation results for the 50% boundary. For *pombe* we set $L = 2 \mu m$ and

$v = 3 \times 10^{-3} \mu\text{m}/\text{s}$ as in Gerganova et al. and find that $D < 0.006 \mu\text{m}^2/\text{s}$ or $k_{\text{off}} < 0.005 \text{s}^{-1}$ result in depletion. To connect these two limiting values we assume a form for $D = (1 - \tau k_{\text{off}})/(\tau C)$ where τ and C are fixed by the limiting values. For *N. crassa* we set $L = 5 \mu\text{m}$ and determine $v = 0.1604 \mu\text{m}/\text{s}$ which is the expected flow rate due to exocytosis at an arclength of $1.589 \mu\text{m}$. For *N. crassa*, $D < 0.802 \mu\text{m}^2/\text{s}$ or $k_{\text{off}} < 0.107 \text{s}^{-1}$ leads to depletion. While the boundary shifts to larger D and k_{off} values for *N. crassa*, this analysis shows that a dissociation rate similar to that of Cdc42-GDP (0.17s^{-1}) would provide sufficient mobility to withstand the membrane flow. We have added this analysis to the manuscript.

Figure R9. Depletion boundaries for *S. pombe* and *N. crassa* determined by either particle-based simulation (lighter lines) or from the dimensional analysis given in Gerganova et al. Sci Adv 2021.

4. In the Methods section on how they did the averaging to get particle concentration profiles, they refer to the “largest cluster” of Cdc42, and to the changing calculation of where the centroid is. It would be good to know how many clusters of Cdc42 are produced, and with what distribution of sizes and lifetimes. Does the model usually have an obvious dominant cluster with rare satellites, or are multi-cluster states common? Also, how stable is the cluster centroid position? If there is usually an obvious dominant cluster, on what timescale does the centroid change position, and by how much?

To address this question, we measured the number of clusters and their sizes as a function of time using a cluster cutoff distance of $0.7 \mu\text{m}$ (Fig. R10). The cluster cutoff distance is the maximum arclength distance two Cdc42-GTP particles can be separated by and still be considered as part of the same cluster. Fig. R10 shows there is always a single large cluster after steady state has been reached for Cdc42-WT and Cdc42-1ritC. For Cdc42-2ritC, a more diffuse patch forms at steady state that transiently breaks up into separate patches and reforms. For Cdc42-3ritC there are several small zones of activation with no dominant patch region at steady state. Therefore, as Cdc42-GDP mobility slows, the number of clusters of active Cdc42 increases.

Figure R10. Cluster sizes over time for WT and ritC conditions, where color indicates the number of clusters at time t , for the simulations of Figures 1E, 3A and Movies S1-S4. We used a cluster cutoff of $0.7 \mu\text{m}$ (i.e. particles within an arclength of $0.7 \mu\text{m}$ from each other are considered to be in the same cluster) and ignored any clusters with fewer than five particles of Cdc42-GTP.

For the same cluster definition, we also calculated the root mean square displacement (RMSD) of the largest Cdc42-GTP patch centroid for the WT and ritC conditions (Fig. R11, left). This shows that the WT, 1ritC, and 2ritC patches remain constrained to a region of the full sphere surface throughout the simulation. This constraining effect appears due to the presence of sGAP since in WT simulations without sGAP the RMSD value increases more quickly (Fig. R11B). We note that the 3ritC patch fluctuates more wildly and has much higher values of RMSD, indicating hotspots changing position on the order of hundreds of seconds by around $2 \mu\text{m}$, though still remaining constrained to approximately half the sphere surface even after 10^4 s (Fig. R11). The exact value of these displacements depends on the cluster cutoff distance used, but the behavior described is independent of this choice.

Figure R11. RMSD, averaged over each starting time t_0 for each time lag τ , of the centroid of the largest Cdc42-GTP cluster for WT and XritC conditions. Averaging started after patch formed at least 5500 s after the initial random placement of particles on sphere surface. Panel A show reference case for WT and ritC conditions with sGAP. Panel B shows the WT case without sGAP and with k_{hydro} increased to 1.75 s^{-1} to allow a single, well-defined Cdc42-GTP patch to form (see Figure R8). Dashed blue line is the average distance between two random points on the surface of a sphere with radius $1.8 \mu\text{m}$.

5. There seemed to be contradictory statements on how Cdc42-GTP diffusion and detachment were modeled for the ritC constructs. In the Results it says “we assumed that the ...Cdc42-GTP...parameters are limited by the ritC anchor and set them the same as for Cdc42-GDP”. This would seem to imply that GTP- and GDP-bound forms now share the same mobility. But in the Methods it says they “lowered the corresponding values of Cdc42-GTP ... so that Cdc42-GDP would always have a greater than or equal mobility compared to Cdc42-GTP”. I don’t understand how both statements can be true. I would assume the Methods version is correct?

The Methods version is the correct statement. We apologize for the lack of clarity in the Results section version – the statement cited above was meant to just focus on the ritC constructs. For all three ritC constructs the diffusion coefficient for Cdc42-GDP, D_D , is lower than the diffusion coefficient for Cdc42-GTP in the WT case, $D_{T,WT}$, so that the expression in the Methods section will simplify to $D_T = \min\{D_D, D_{T,WT}\} = D_D$. The reasoning is similar concerning the unbinding rate for Cdc42-2ritC and Cdc42-3ritC, where r_D is lower than $r_{T,WT}$ so that $r_T = \min\{r_D, r_{T,WT}\} = r_D$. For Cdc42-1ritC the unbinding rate, r_D , is higher than $r_{T,WT}$ so that $r_T = \min\{r_D, r_{T,WT}\} = r_{T,WT}$ and so the GTP and GDP bound forms have unique unbinding rates in this case. We updated this sentence in the Results section to be clearer as follows:

“The two slower Cdc42-ritC constructs (Cdc42-2ritC and Cdc42-3ritC) have lower Cdc42-GDP diffusion and dissociation constants than WT Cdc42-GTP (Figures 1D, 2B, S3); thus, for the 2ritC and 3ritC cases we assumed that the corresponding mobility parameters of the GTP-bound form, in association or not with Scd1/2, are limited by the ritC anchor and set them the same as for the GDP-bound form.”

6. For the simulations of diploid cells, particle numbers were doubled. But was the sphere volume also doubled? If not, aren't these simulations of overexpressing cells rather than diploids?

We thank the reviewer for pointing out this discrepancy. Indeed, our prior “diploid” simulations had doubled particle numbers but were on the same sized sphere. We fixed this issue by doubling the volume as well as the particle number. The qualitative result that having a WT copy rescues patch formation remain unchanged and the distribution of Cdc42-XritC matches the experimentally observed ones with depletion of Cdc42-3ritC. We display these new results in the updated Supplemental Figure S5.

7. It was not clear which combinations of GAP deletions (other than the triple) were able to suppress the 3x ritC Cdc42 lethality. Do single GAP deletes suppress? If so, which ones? Do they look any better than the triple? Simulation of GAP deletion cells is described only for the sGAP: what happens if one lowers khydro instead?

We were also very interested in finding out whether other combinations of GAP deletions would suppress lethality of slow Cdc42. Our experiment was to sporulate heterozygous mutants for all GAP and Cdc42 loci, which could in principle yield any combination of mutants, and so we were hoping to find all the viable combinations. In this experiment, when genotyping the viable colonies obtained from germinated spores, we initially recovered other combinations than the triple GAP mutant. However, in further PCR tests to verify genotypes, the single/double mutants contained not only the *cdc42-3ritC* slow allele but also WT *cdc42* coding sequence, suggesting that they had arisen from aberrant meiotic figures. The triple mutants contained only the *cdc42-3ritC* slow allele. While this suggests that other combinations of GAP mutants do not support viability with the slow *cdc42-3ritC*, we prefer to remain careful with this negative result and abstain from making strong statements in the paper.

To be certain about the viable *cdc42-3ritC* triple GAP mutants, we also obtained them from tetrad dissections, and these are the alleles we used to show morphology in Figure 4C.

We also performed simulations to check the effect of reducing khydro in simulations. We find that reduction of khydro does lead to an increase in Cdc42-GTP together with a narrowing of the patch. However, for our reference WT parameters, this effect is weaker than that of reducing sGAP by 50%. It

is also not sufficient to bring active cdc42-3ritC cells above the survival threshold (dashed line below in Figure R12 and updated in the main Figure 4A).

Figure R13. Updated Figure 4A that includes the case where k_{hydro} is reduced to 1/100th of its reference value. This does increase the amount of Cdc42-GTP on the surface, but the effect is not as strong as reducing the amount of sGAP.

8. There is a typo in the Exo/endocytosis Methods where it says “wexo or wexo, respectively”. Presumably one of those is intended to be wendo.

Thanks - corrected!

REVIEWERS' COMMENTS

Reviewer #1 (Remarks to the Author):

We recommend publication.